# Low-Regret Active Learning

## Abstract

We develop an online learning algorithm for identifying unlabeled data points that are most informative for training (i.e., active learning). By formulating the active learning problem as the prediction with sleeping experts problem, we provide a regret minimization framework for identifying relevant data with respect to any given definition of informativeness. Motivated by the successes of ensembles in active learning, we define regret with respect to an omnipotent algorithm that has access to an infinity large ensemble. At the core of our work is an efficient algorithm for sleeping experts that is tailored to achieve low regret on easy instances while remaining resilient to adversarial ones. Low regret implies that we can be provably competitive with an ensemble method *without the computational burden of having to train an ensemble.* This stands in contrast to state-of-the-art active learning methods that are overwhelmingly based on greedy selection, and hence cannot ensure good performance across problem instances with high amounts of noise. We present empirical results demonstrating that our method (i) instantiated with an informativeness measure consistently outperforms its greedy counterpart and (ii) reliably outperforms uniform sampling on real-world scenarios.

## 1 Introduction

Modern neural networks have been highly successful in a wide variety of applications ranging from Computer Vision (Feng et al., 2019) to Natural Language Processing (Brown et al., 2020). However, these successes have come on the back of training large models on massive labeled data sets, which may be costly or even infeasible to obtain in other applications. For instance, applying deep networks to the task of cancer detection requires medical images that can only be labeled with the expertise of healthcare professionals, and a single accurate annotation may come at the cost of a biopsy on a patient (Shen et al., 2019).

*Active learning* focuses on alleviating the high label-cost of learning by only querying the labels of points that are deemed to be the most informative. The notion of informativeness is not concrete and may be defined in a task-specific way. Unsurprisingly, prior work in active learning has primarily focused on devising proxy metrics to appropriately quantify the informativeness of each data point in a tractable way. Examples include proxies based on model uncertainty (Gal et al., 2017), clustering (Sener & Savarese, 2017; Ash et al., 2019), and margin proximity (Ducoffe & Precioso, 2018) (see (Ren et al., 2020) for a detailed survey).

An overwhelming majority of existing methods are based on greedy selection of the points that are ranked as most informative with respect to the proxy criterion. Despite the intuitiveness of this approach, it is known to be highly sensitive to outliers and to training noise, and observed to perform significantly worse than uniform sampling on certain tasks (Ebrahimi et al., 2020) – as Fig. 1 also depicts. In fact, this shortcoming manifests itself even on reportedly redundant data sets, such as MNIST, where existing approaches can lead to models with up to 15% (absolute terms) higher test error (Muthakana, 2019) than those obtained with uniform sampling. In sum, the general lack of robustness and reliability of prior (greedy) approaches impedes their widespread applicability to high-impact deep learning tasks.

In this paper, we propose a low-regret active learning framework and develop an algorithm that can be applied with any user-specified notion of informativeness. Our approach deviates from the standard greedy paradigm and instead formulates the active learning problem as that of learning with expert advice in an adversarial environment. Motivated by the widespread success of ensemble approaches in active learning (Beluch et al.,

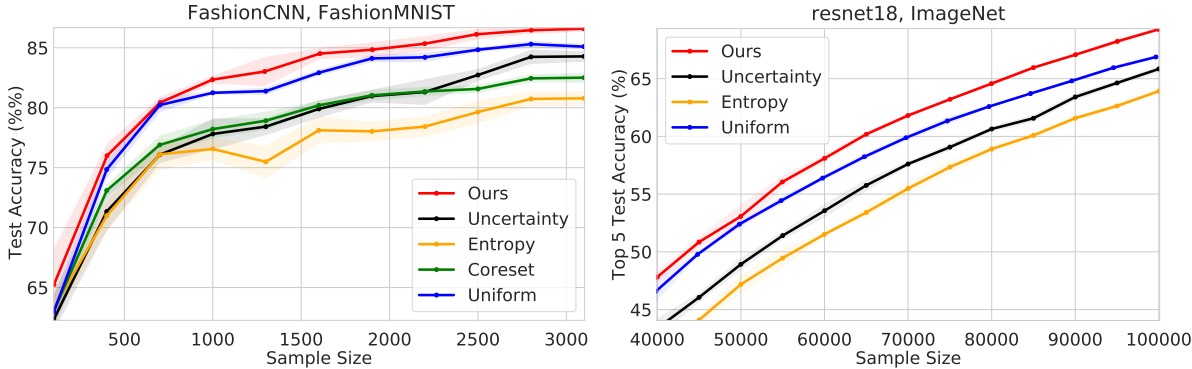

Figure 1: Evaluations on FashionMNIST and ImageNet with benchmark active learning algorithms. Existing approaches based on greedy selection are not robust and may perform significantly worse than uniform sampling.

2018), we define regret with respect to an oracle approach that has access to an infinitely large model ensemble. This oracle represents an omniscient algorithm that can completely smoothen out all training noise and compute the expected informativeness of data points over the randomness in the training. *Low regret in this context (roughly) implies that we are robust to training noise and provably competitive with the performance of an ensemble, without the computational burden of having to train an ensemble of models.*

Overall, our work aims to advance the development of efficient and robust active learning strategies that can be widely applied to modern deep learning tasks. In particular, we:

1. Formulate active learning as a prediction with sleeping experts problem and develop an efficient, predictive algorithm for low-regret active learning,

2. Establish an upper bound on the expected regret of our algorithm that scales with the difficulty of the problem instance,

3. Compare and demonstrate the effectiveness of the presented method on a diverse set of benchmarks, and present its uniformly superior performance over competitors across varying scenarios.

## 2 Background & Problem Formulation

We consider the setting where we are given a set of $n$ unlabeled data points $\mathcal{P} \subset \mathcal{X}^n$ from the input space $\mathcal{X} \subset \mathbb{R}^d$. We assume that there is an oracle ORACLE that maps each point $x \in \mathcal{P}$ to one of $k$ categories. Given a network architecture and sampling budget $b \in \mathbb{N}_+$, our goal is to generate a subset of points $\mathcal{S} \subset \mathcal{P}$ with $|\mathcal{S}| = b$ such that training on $\{(x, \text{ORACLE}(x))_{x \in \mathcal{S}}\}$ leads to the most accurate model $\theta$ among all other choices for a subset $\mathcal{S} \subset \mathcal{P}$ of size $b$.

The iterative variant of acquisition procedure is shown as Alg. 1, where ACQUIRE is an active learning algorithm that identifies (by using $\theta_{t-1}$) $b_t$ unlabeled points to label at each iteration $t \in [T]$ and TRAIN trains a model initialized with $\theta_{t-1}$ using the labeled set of points. We emphasize that prior work has overwhelmingly used the SCRATCH option (Line 6, Alg. 1), which entails discarding the model information $\theta_{t-1}$ from the previous iteration and training a randomly initialized model from scratch on the set of labeled points acquired thus far, $\mathcal{S}$.

**Active Learning** Consider an *informativeness function* $g : \mathcal{X} \times \Theta \to [0, 1]$ that quantifies the informativeness of each point $x \in \mathcal{X}$ with respect to the model $\theta \in \Theta$, where $\Theta$ is the set of all possible parameters for the given architecture. An example of the gain function is the maximum variation ratio (also called the *uncertainty* metric) defined as $g(x, \theta) = 1 - \max_{i \in [k]} f_\theta(x)_i$, where $f_\theta(x) \in \mathbb{R}^k$ is the softmax output of the model $\theta$ given input $x$. As examples, the *gain* $g(x, \theta)$ of point $x$ is 0 if the network is absolutely certain about the label of $x$ and $1 - 1/k$ when the network's prediction is uniform. In the context of Alg. 1, prior work on active learning (Muthakana, 2019; Geifman & El-Yaniv, 2017; Gal et al., 2017; Sener & Savarese, 2017) has

---

**Algorithm 1** ACTIVELEARNING

---

**Input:** Set of points $\mathcal{P} \subseteq \mathbb{R}^{d \times n}$, ACQUIRE: an active learning algorithm for selecting labeled points

1: $\mathcal{S} \leftarrow \emptyset$; $\theta_0 \leftarrow$ Randomly initialized network model;
2: **for** $t \in [T] = \{1, \ldots, T\}$ **do**
3:     $\mathcal{S}_t \leftarrow$ ACQUIRE$(\mathcal{P} \setminus \mathcal{S}, b_t, \theta_{t-1})$ {Get new batch of $b_t \in \mathbb{N}_+$ points to label using algorithm ACQUIRE}
4:     $\mathcal{S} \leftarrow \mathcal{S} \cup \mathcal{S}_t$ {Add new points}
5:     (if SCRATCH option) $\theta_{t-1} \leftarrow$ Randomly initialized network
6:     $\theta_t \leftarrow$ TRAIN$(\theta_{t-1}, \{(x, \text{ORACLE}(x))_{x \in \mathcal{S}}\})$ {Train network on the labeled samples thus far}
7: **end for**
8: **return** $\theta_T$

---

generally focused on greedy acquisition strategies (ACQUIRE in Alg. 1) that rank the remaining unlabeled points by their informativeness $g(x, \theta_{t-1})$ *as a function of the model $\theta_{t-1}$*, and pick the top $b_t$ points to label.

## 2.1 Greedy's Shortcoming & Ensembles

As observed in prior work (Ebrahimi et al., 2020; Muthakana, 2019) and seen in our evaluations, e.g., Fig. 1, greedy approaches may perform significantly worse than naive uniform sampling. To understand why this could be happening, note that at iteration $t \in [T]$ the greedy approach makes a judgment about the informativeness of each point using only the model $\theta_{t-1}$ (Line 4 of Alg. 1). However, in the deep learning setting where stochastic elements such as random initialization, stochastic optimization, (randomized) data augmentation, and dropout are commonly present, $\theta_{t-1}$ is itself a random variable with non-negligible variance. This means that, for example, we could get unlucky with our training and obtain a deceptive model $\theta_{t-1}$ (e.g., training diverged) that assigns high gains (informativeness) to points that may not truly be helpful towards training a better model. Nevertheless, GREEDY would still base the entirety of the decision making solely on $\theta_{t-1}$ and blindly pick the top-$b_t$ points ranked using $\theta_{t-1}$, leading to a misguided selection. This also applies to greedy clustering, e.g., CORESET (Sener & Savarese, 2017), BADGE (Ash et al., 2019).

Relative to GREEDY, the advantage of ensemble methods is that they are able to smoothen out the training noise and select points with *high expected informativeness* over the randomness in the training. In other words, rather than greedily choosing the points with high informativeness $g(x, \theta_{t-1})$ with respect to a single model, ensembles can be viewed as selecting points with respect to a finite-sample approximation of $\mathbb{E}_{\theta_{t-1}}[g(x, \theta_{t-1})]$ by considering the informativeness over multiple trained models.

## 2.2 Active Learning as Prediction with Expert Advice

Roughly, we consider our active learning objective to be the selection of points with maximum *expected* informativeness $\mathbb{E}_{\theta_{t-1}}[g(x, \theta_{t-1})]$ over the course of $T$ active learning iterations. By doing so, we aim to smoothen out the training noise in evaluating the informativeness as ensembles do, but *in an efficient way by training only a single model* at a time as in Alg. 1. For sake of simplicity, we present the problem formulation for the case of sampling a single data point at each iteration rather than a batch. The generalization of this problem to batch sampling and the corresponding algorithm and analysis are in Sec. 4.2 and the Appendix (Sec. B).

**Notation** To formalize this objective, we let $g_{t,i}(\theta_{t-1})$ denote the gain of the $i^{\text{th}}$ point in $\mathcal{P}$ in round $t \in [T]$ with respect to $\theta_{t-1}$. Rather than seeking to optimize gains, we follow standard convention in online learning (Orabona, 2019) and consider the minimization of losses $\ell_{t,i}(\theta_{t-1}) = 1 - g_{t,i}(\theta_{t-1}) \in [0, 1]$ instead. We let $\xi_0, \ldots, \xi_T$ denote the training noise at each round $t$. For any given realization of a subset of points $\mathcal{S} \subset \mathcal{P}$ and noise $\xi$, define $\mathcal{M}(\mathcal{S}, \xi)$ to be the deterministic function that returns the trained model $\theta$ on the set $\mathcal{S}$. In the context of Alg. 1, the model $\theta_t$ at each iteration is given by $\theta_t = \mathcal{M}(\cup_{\tau=1}^{t} \mathcal{S}_\tau, \xi_t)$, The loss at each time step can now be defined more rigorously as $\ell_t(\theta_{t-1}) = \ell_t(\mathcal{M}(\cup_{\tau=1}^{t-1} \mathcal{S}_\tau, \xi_{t-1}))$. We will frequently abbreviate the loss vector at round $t$ as $\ell_t(\xi_{t-1})$ to emphasize the randomness over the training noise or simply as $\ell_t$ with the understanding that it is a function of the random variables $\mathcal{S}_1, \ldots, \mathcal{S}_{t-1}$ and $\xi_{t-1}$.

Since the expectation cannot be computed exactly a priori knowledge about the problem, we turn to an online learning strategy and aim to minimize the *regret* over $T$ active learning iterations. More specifically, we follow the learning with prediction advice formulation where each data point (regardless of whether it has already been labeled) is considered an expert. The main idea is to pick the most informative data points, or experts in this context. At each iteration $t$, rather than picking a points to label in a deterministic way as do most greedy strategies, we propose using a probability distribution $p_t \in \Delta_t$ to sample the points instead, where $\Delta = \{p \in [0,1]^n : \sum_{j=1}^n p_j = 1\}$. For the filtration $\mathcal{F}_t = \sigma(\xi_0, \mathcal{S}_1, \ldots, \xi_{t-1}, \mathcal{S}_t)$ with $|\mathcal{S}_t| = 1$ for all $t$, note that the conditional expected loss at each iteration is $\mathbb{E}_{\mathcal{S}_t, \xi_{t-1}}[\ell_{t, \mathcal{S}_t}(\xi_{t-1}) | \mathcal{F}_{t-1}] = \langle p_t, \mathbb{E}_{\xi_{t-1}}[\ell_t(\xi_{t-1})] \rangle$ since $p_t$ is independent of $\xi_{t-1}$.

Under this setting a natural first attempt at a formulation of regret is to define it as in the problem of learning with expert advice. To this end, we define the instantaneous regret $r_{t,i}$ to measure the expected loss under the sampling distribution $p_t$ relative to that of picking the $i^{\text{th}}$ point for a given realization of $\ell_t$, i.e.,

$$r_{t,i} = \langle p_t, \ell_t(\xi_{t-1}) \rangle - \ell_{t,i}(\xi_{t-1}).$$

However, the sampling method and the definition of regret above are not well-suited for the problem of active learning because (i) $p_t$ may sample points that have already been labeled and (ii) the instantaneous regret for those points that are already sampled should be 0 so that we can define appropriately define regret over the unlabeled data points.

**Sleeping Experts and Dynamic Regret**   To resolve these challenges and ensure that we only sample from the pool of unlabeled data points, we generalize the prior formulation to one with *sleeping experts* (Saha et al., 2020; Luo & Schapire, 2015; Gaillard et al., 2014; Kleinberg et al., 2010). More concretely, let $\mathcal{I}_{t,i} \in \{0,1\}$ denote whether expert $i \in [n]$ is sleeping in round $t$. The sleeping expert problem imposes the constraint that $\mathcal{I}_{t,i} = 0 \Rightarrow p_{t,i} = 0$. For the data acquisition setting, we define $\mathcal{I}_{t,i} = \mathbb{1}\{x_i$ has not yet been labeled$\}$, so that we do not sample already-labeled points. Then, the definition of instantenous regret becomes

$$r_{t,i} = (\langle p_t, \ell_t(\xi_{t-1}) \rangle - \ell_{t,i}(\xi_{t-1}))\mathcal{I}_{t,i},$$

and the regret over $T$ iterations with respect to a competing sequence of samples $i_1^*, \ldots, i_T^* \in [n]$ is defined as

$$\mathcal{R}(i_{1:T}^*) = \sum_{t=1}^T \mathbb{E}_{\xi_{t-1}}[r_{t,i_t^*}] = \sum_{t=1}^T \left( \langle p_t, \mathbb{E}_{\xi_{t-1}}[\ell_t(\xi_{t-1})] \rangle - \mathbb{E}_{\xi_{t-1}}[\ell_{t,i_t^*}(\xi_{t-1})] \right) \mathcal{I}_{t,i_t^*} \tag{1}$$

Our overarching goal is to minimize the maximum expected regret, which is at times referred to as the dynamic pseudoregret (Wei et al., 2017), relative to any sequence of samples $i_1, \ldots, i_T \in [n]$

$$\max_{(i_t)_{t \in [T]}} \mathbb{E}[\mathcal{R}(i_{1:T})].$$

## 3   Method

In this section we motivate and present Alg. 2, an efficient online learning algorithm with instance-dependent guarantees that performs well on predictable sequences while remaining resilient to adversarial ones. Additional implementation details are outlined in Sec. C of the supplementary.

### 3.1   Background

Algorithms for the prediction with sleeping experts problem have been extensively studied in literature (Gaillard et al., 2014; Luo & Schapire, 2015; Saha et al., 2020; Kleinberg et al., 2010; Shayestehmanesh et al., 2019; Koolen & Van Erven, 2015). These algorithms enjoy strong guarantees in the adversarial setting; however, they suffer from (i) sub-optimal regret bounds in predictable settings and/or (ii) high computational complexity. Our approach hinges on the observation that the active learning setting may not always be adversarial in practice, and if this is the case, we should be competitive with greedy approaches. For example, we may expect the informativeness of the points to resemble a predictable sequence plus random noise which models

the random components of the training (see Sec. 2) at each time step. This (potential) predictability in the corresponding losses motivates an algorithm that can leverage predictions about the loss for the next time step to achieve lower regret by being more aggressive – akin to GREEDY – when the losses do not vary significantly over time.

## 3.2 AdaProd$^+$

To this end, we extend the Optimistic Adapt-ML-Prod algorithm (Wei et al., 2017) (henceforth, OAMLProd) to the active learning setting with batched plays where the set of experts (unlabeled data points) is changing and/or unknown in advance. Optimistic online learning algorithms are capable of incorporating predictions $\hat{\ell}_{t+1}$ for the loss in the next round $\ell_{t+1}$ and guaranteeing regret as a function of the predictions' accuracy, i.e., as a function of $\sum_{t=1}^{T} ||\ell_t - \hat{\ell}_t||_\infty^2$. Although we could have attempted to extend other optimistic approaches (Steinhardt & Liang, 2014; Orabona, 2019; Mohri & Yang, 2015; Rakhlin & Sridharan, 2013), the work of (Wei et al., 2017) ensures – to the best of our knowledge – the smallest regret in predictable environments when compared to related approaches.

---

**Algorithm 2** ADAPROD$^+$

1: For all $i \in [n]$, initialize $R_{1,i} \leftarrow 0$; $C_{1,i} \leftarrow 0$; $\eta_{0,(1,i)} \leftarrow \sqrt{\log n}$; $w_{0,(1,i)} = 1$; $\hat{r}_{1,i} = 0$;
2: **for** each round $t \in [T]$ **do**
3:      $\mathcal{A}_t \leftarrow \{i \in [n] : \mathcal{I}_{t,i} = 1\}$ {Set of awake experts, i.e., set of unlabeled data points}
4:      $p_{t,i} \leftarrow \sum_{s \in [t]} \eta_{t-1,(s,i)} w_{t-1,(s,i)} \exp(\eta_{t-1,(s,i)} \hat{r}_{t,i})$     for each $i \in \mathcal{A}_t$
5:      $p_{t,i} \leftarrow p_{t,i} / \sum_{j \in \mathcal{A}_t} p_{t,j}$ for each $i \in \mathcal{A}_t$ {Normalize}
6:      Adversary reveals $\ell_t$ and we suffer loss $\tilde{\ell}_t = \langle \ell_t, p_t \rangle$
7:      For all $i \in \mathcal{A}_t$, $r_{t,i} \leftarrow \tilde{\ell}_t - \ell_{t,i}$ and $C_{t,i} \leftarrow 0$
8:      For all $i \in \mathcal{A}_t$ and $s \in [t]$, set $C_{s,i} \leftarrow C_{s,i} + (\hat{r}_{t,i} - r_{t,i})^2$
9:      Get prediction $\hat{r}_{t+1} \in [-1, 1]^n$ for next round (see Sec. 3.2)
10:      For all $i \in \mathcal{A}_t$, set $w_{t-1,(t,i)} \leftarrow 1$, $\eta_{t-1,(t,i)} \leftarrow \sqrt{\log n}$, and for all $s \in [t]$, set

$$\eta_{t,(s,i)} \leftarrow \min \left\{ \eta_{t-1,(s,i)}, \frac{2}{3(1 + \hat{r}_{t+1,i})}, \sqrt{\frac{2 \log(n)}{C_{s,i}}} \right\} \quad \text{and}$$

$$w_{t,(s,i)} \leftarrow \left( w_{t-1,(s,i)} \exp \left( \eta_{t-1,(s,i)} r_{t,i} - \eta_{t-1,(s,i)}^2 (r_{t,i} - \hat{r}_{t,i})^2 \right) \right)^{\eta_{t,(s,i)} / \eta_{t-1,(s,i)}}$$

11: **end for**

---

Our algorithm ADAPROD$^+$ is shown as Alg. 2. Besides its improved computational efficiency relative to OAMLProd in the active learning setting, ADAPROD$^+$ is also the result of a tightened analysis that leads to significant practical improvements over OAMLProd as shown in Fig. 5 of Sec. 5.5. Our insight is that our predictions can be leveraged to improve practical performance by allowing larger learning rates to be used without sacrificing theoretical guarantees (Line 10 of Alg. 2). Empirical comparisons with Adapt-ML-Prod and other state-of-the-art algorithms can be found in Sec. 5.5.

**Generating Predictions** Our approach can be used with general predictors $\hat{\ell}_t$ for the true loss $\ell_t$ at round $t$, however, to obtain bounds in terms of the temporal variation in the losses, we use the most recently observed loss as our prediction for the next round, i.e., $\hat{\ell}_t = \ell_{t-1}$. A subtle issue is that our algorithm requires a prediction $\hat{r}_t \in [-1, 1]^n$ for the instantaneous regret at round $t$, i.e., $\hat{r}_t = \langle p_t, \hat{\ell}_t \rangle - \hat{\ell}_t$, which is not available since $p_t$ is a function of $r_t$. To achieve this, we follow (Wei et al., 2017) and define the mapping $\hat{r}_t : \alpha \mapsto (\alpha - \hat{\ell}_t) \in [-1, 1]^n$ and perform a binary search over the update rule in Lines 4-5 of Alg. 2 so that $\alpha \in [0, 1]$ is such that $\alpha = \langle p_t(\hat{r}_t(\alpha)), \hat{\ell}_t \rangle$, where $p_t(\hat{r}_t(\alpha))$ is the distribution obtained when $\hat{r}_t(\alpha)$ is used as the optimistic prediction in Lines 4-5. The existence of such an $\alpha$ follows by applying the intermediate value theorem to the continuous update.

### 3.3 Back to Active Learning

To unify ADAPROD$^+$ with Alg. 1, observe that we can define the ACQUIRE function to be a procedure that at time step $t$ first samples a point by sampling with respect to probabilities $p_t$, obtains the (user-specified) losses $\ell_t$ with respect to the model $\theta_{t-1}$, and passes them to Alg. 2 as if they were obtained from the adversary (Line 6). This generates an updated probability distribution $p_{t+1}$ and we iterate.

To generalize this approach to sampling a batch of $b_t$ points, we build on ideas from (Uchiya et al., 2010). Here, we provide an outline of this procedure; the full details along with the code are provided in the supplementary (Sec. B). At time $t$, we apply a capping algorithm (Uchiya et al., 2010) to the probability $p_t$ generated by ADAPROD$^+$ – which takes $\mathcal{O}(n_t \log n_t)$ time, where $n_t \leq n$ is the number of remaining unlabeled points at iteration $t$ – to obtain a modified distribution $\tilde{p}_t$ satisfying $\max_i \tilde{p}_{t,i} \leq 1/b_t$. This projection to the capped simplex ensures that the scaled version of $\tilde{p}_t$, $\hat{p}_{t,i} = b_t \tilde{p}_{t,i}$, satisfies $\hat{p}_{t,i} \in [0,1]$ and $\sum_j \hat{p}_{t,j} = b_t$. Now the challenge is to sample exactly $b_t$ distinct points according to probability $\hat{p}_t$. To achieve this, we use a dependent randomized rounding scheme (Gandhi et al., 2006) (Alg. 4 in supplementary) that runs in $\mathcal{O}(n_t)$ time. The overall computational overhead of batch sampling is $\mathcal{O}(n_t \log n_t)$.

### 3.4 Flexibility via Proprietary Loss

We end this section by underscoring the generality of our approach, which can be applied *off-the-shelf with any definition of informativeness measure that defines the loss $\ell \in [0,1]^n$*, i.e., 1 - informativeness. For example, our framework can be applied with the *uncertainty metric* as defined in Sec. 2 by defining the losses to be $\ell_{t,i} = \max_{j \in [k]} f_{\theta_{t-1}}(x_i)_j$. As we show in Sec. 5.4, we can also use other popular notions of informativeness such as Entropy (Ren et al., 2020) and the BALD metrics (Gal et al., 2017) to obtain improved results *relative to greedy selection*. This flexibility means that our approach can always be instantiated with any state-of-the-art notion of informativeness, and consequently, can scale with future advances in appropriate notions of informativeness widely studied in literature.

## 4 Analysis

In this section, we present the theoretical guarantees of our algorithm in the learning with sleeping experts setting. Our main result is an instance-dependent bound on the dynamic regret of our approach in the active learning setting. We focus on the key ideas in this section and refer the reader to the Sec. A of the supplementary for the full proofs and generalization to the batch setting.

The main idea of our analysis is to show that ADAPROD$^+$ (Alg. 2), which builds on Optimistic Adapt-ML-Prod (Wei et al., 2017), retains the adaptive regret guarantees of the time-varying variant of their algorithm without having to know the number of experts a priori (Wei et al., 2017). Inspired by AdaNormalHedge (Luo & Schapire, 2015), we show that our algorithm can efficiently ensure adaptive regret by keeping track of $\sum_{t \in 1}^{T} n_t \leq nt$ experts at time step $t$, where $n_t$ denotes the number of unlabeled points remaining, $n_t = \sum_{i=1}^{n} \mathcal{I}_{t,i}$, rather than $nt$ experts as in prior work. This leads to efficient updates and applicability to the active learning setting where the set of unlabeled points remaining (experts) significantly shrinks over time.

Our second contribution is an improved learning rate schedule (Line 10 of Alg. 2) that arises from a tightened analysis that enables us to get away with strictly larger learning rates without sacrificing any of the theoretical guarantees. For comparison, the learning rate schedule of (Wei et al., 2017) would be $\eta_{t,(s,i)} = \min\{1/4, \sqrt{2\log(n)/(1 + C_{s,i})}\}$ in the context of Alg. 2. It turns out that the dampening factor of 1 from the denominator can be removed, and the upper bound of $1/4$ is overly-conservative and can instead be replaced by $\min\{\eta_{t-1,(s,i)}, 2/(3(1 + \hat{r}_{t+1,i}))\}$. This means that we can leverage the predictions at round $t$ to set the threshold in a more informed way. Although this change does not improve (or change) the worst-case regret bound asymptotically, our results in Sec. 5 (see Fig. 5) show that it translates to significant practical improvements in the active learning setting.

### 4.1 Point Sampling

We highlight the two main results here and refer to the supplementary for the full analysis and proofs. The lemma below bounds the *adaptive regret* of ADAPROD$^+$, which concerns the cumulative regret over a time interval $[t_1, t_2]$, with respect to $C_{t_2,(t_1,i)} = \sum_{t=t_1}^{t_2} (r_{t,i} - \hat{r}_{t,i})^2$.

**Lemma 1** (Adaptive Regret of ADAPROD$^+$). *For any $t_1 \leq t_2$ and $i \in [n]$, Alg. 2 ensures that*

$$\sum_{t=t_1}^{t_2} r_{t,i} \leq \mathcal{O}\left(\log n + \log\log n + (\sqrt{\log n} + \log\log n)\sqrt{C_{t_2,(t_1,i)}}\right),$$

*where $C_{t_2,(t_1,i)} = \sum_{t=t_1}^{t_2} (r_{t,i} - \hat{r}_{t,i})^2$ and $r_{t,i} = (\langle \ell_t, p_t \rangle - \ell_{t,i})\mathcal{I}_{t,i}$ is the instantaneous regret of $i \in [n]$ at time $t$ and $\hat{r}_{t,i} = (\langle \hat{\ell}_t, p_t \rangle - \hat{\ell}_{t,i})\mathcal{I}_{t,i}$ is the predicted instantenous regret as a function of the optimistic loss vector $\hat{\ell}_t$ prediction.*

It turns out that there is a deep connection between dynamic and adaptive regret, and that an adaptive regret bound implies a dynamic regret bound (Luo & Schapire, 2015; Wei et al., 2017). The next theorem follows by an invocation of Lemma 1 to multiple ($\mathcal{O}(T/B)$) time blocks of length $B$ and additional machinery to bound the regret per time block. The bound on the expected dynamic regret is a function of the sum of the prediction error of $\hat{r}_t$ which is a function of our loss prediction $\ell_t$, and $\mathcal{D}_T$, which is the drift in the *expected regret*

$$\mathcal{V}_T = \sum_{t \in [T]} \|r_t - \hat{r}_t\|_\infty^2 \quad \text{and} \quad \mathcal{D}_T = \sum_{t \in [T]} \|\mathbb{E}[r_t] - \mathbb{E}[r_{t-1}]\|_\infty,$$

where the expectation $\mathbb{E}[r_t] = \mathbb{E}[(\langle p_t, \ell_t(\xi_{t-1}) \rangle - \ell_t(\xi_{t-1})) \odot \mathcal{I}_t]$ is taken over all the relevant random variables, i.e., the training noise $\xi_{0:t-1}$ and the algorithm's actions up to point $t$, $\mathcal{S}_{1:t}$, with $\odot$ denoting the Hadamard (entry-wise) product. We note that by Hölder's inequality, $\mathcal{V}_T \leq 4\sum_{t \in [T]} \|\ell_t - \hat{\ell}_t\|_\infty^2$, which makes it easier to view the quantity as the error between the loss predictions $\hat{\ell}_t$ and the realized ones $\ell_t$.

**Theorem 2** (Dynamic Regret). *ADAPROD$^+$ takes at most $\tilde{\mathcal{O}}(tn_t)$ [1] time for the $t^{th}$ update and for batch size $b = 1$ for all $t \in [T]$, guarantees that over $T$ steps,*

$$\max_{(i_t)_{t \in [T]}} \mathbb{E}[\mathcal{R}(i_{1:T})] \leq \hat{\mathcal{O}}\left(\sqrt[3]{\mathbb{E}[\mathcal{V}_T]\mathcal{D}_T T \log n} + \sqrt{\mathcal{D}_T T \log n}\right),$$

*where $\hat{\mathcal{O}}(\cdot)$ suppresses $\log T$ factors.*

Note that in scenarios where GREEDY fails, i.e., instances where there is a high amount of training noise, the expected variance of the losses with respect to the training noise $\mathbb{E}[\mathcal{V}_T]$ may be on the order of $T$. However, even in high noise scenarios, the drift in the *expectation* of the regret may be sublinear, e.g., a distribution of losses that is fixed across all iterations $\mathbb{E}[\ell_1] = \cdots = \mathbb{E}[\ell_T]$, i.e., a stationary distribution, but with high variance $\mathbb{E}[\|\ell_t - \mathbb{E}[\ell_t]\|]$. This means that sublinear dynamic regret is possible with ADAPROD$^+$ even in noisy environments, since then $\mathcal{D}_t = o(T)$, $\mathbb{E}[\mathcal{V}_T] = \Theta(T)$ and $\sqrt[3]{\mathbb{E}[\mathcal{V}_T]\mathcal{D}_T T \log n} + \sqrt{\mathcal{D}_T T \log n} = o(T)$.

### 4.2 Batch Sampling

In the previous subsection, we established bounds on the regret in the case where we sampled a single point in each round $t$. Here, we establish bounds on the performance of Alg. 2 with respect to sampling a batch of $b \geq 1$ points $\mathcal{S}_t$ in each round $t$ without any significant modifications to the presented method. To do so, we make the mild assumption that the time horizon $T$, the size of the data set $n$, and the batch size $b$ are so that $\max_{i \in [n]} p_{t,i} \leq 1/b$ for all $t \in [T]$. We can then define the scaled probabilities $\rho_{t,i} = bp_{t,i}$ and sample according to $\rho_{t,i}$. As detailed in the Appendix (Sec. B), there is a linear-time randomized rounding scheme for picking exactly $b$ samples so that each sample is picked with probability $\rho_{t,i}$. Note that $\rho_{t,i} \in [0,1]$ for all $t$ and $i \in [n]$ by the assumption.

---

[1] We use $\tilde{\mathcal{O}}(\cdot)$ to suppress $\log T$ and $\log n$ factors.

For batch sampling, we override the definition of $r_{t,i}$ and define it with respect to the sampling distribution $\rho$ so that

$$r_{t,i} = \left( \frac{\langle \rho_t, \ell_t(\xi_{t-1}) \rangle}{b} - \ell_{t,i}(\xi_{t-1}) \right) \mathcal{I}_{t,i}.$$

Now let $\mathcal{S}_1^*, \ldots, \mathcal{S}_T^*$ be a competing sequence of samples such that $|\mathcal{S}_t^*| = b$. Then, the expected regret with respect to our samples $\mathcal{S}_t \sim \rho_t$ over $T$ iterations is expressed by.

$$\mathcal{R}(\mathcal{S}_{1:T}^*) = \sum_{t=1}^{T} \sum_{j=1}^{b} \mathbb{E}_{\xi_{t-1}}[r_{t,\mathcal{S}_{tj}^*}],$$

where $\mathcal{S}_{tj}^* \in [n]$ denotes the $j^{\text{th}}$ sample of the competitor at step $t$. The next theorem generalizes Theorem 2 to batch sampling, which comes at the cost of a factor of $b$ in the regret.

**Theorem 3** (Dynamic Regret)**.** ADAPROD$^+$ *with batch sampling of b points guarantees that over T steps,*

$$\max_{(\mathcal{S}_t)_{t \in [T]} : |\mathcal{S}_t| = b \, \forall t} \mathbb{E}\left[ \mathcal{R}(\mathcal{S}_{1:T}) \right] \leq \hat{\mathcal{O}} \left( b \sqrt[3]{\mathbb{E}\left[ \mathcal{V}_T \right] \mathcal{D}_T T \log n} + b \sqrt{\mathcal{D}_T T \log n} \right).$$

We note that the assumption imposed in this subsection is not restrictive in the context of active learning where we assume the pool of unlabeled samples is large and that we only label a small batch at a time, i.e., $n \gg b$. In our experimental evaluations, we verified that it held for all the scenarios and configurations presented in this paper (Sec. 5 and Sec. C of the appendix). Additionally, by our sleeping expert formulation, as soon as the probability of picking a point $p_{t,i}$ starts to become concentrated, we will sample point $i$ and set $p_{t,i} = 0$ with very high probability. Hence, our algorithm inherently discourages the concentration of the probabilities at any one point. Relaxing this assumption rigorously is an avenue for future work.

## 5 Results

In this section, we present evaluations of our algorithm and compare the performance of its variants on common vision tasks. The full set of results and our codebase be found in the supplementary material (Sec. C). Our evaluations across a diverse set of configurations and benchmarks demonstrate the practical effectiveness and reliability of our method. In particular, they show that our approach (i) is the only one to significantly improve on the performance of uniform sampling across all scenarios, (ii) reliably outperforms competing approaches even with the intuitive UNCERTAINTY metric (Fig. 2,3), (iii) when instantiated with other metrics, leads to strict improvements over greedy selection (Fig. 4), and (iv) outperforms modern algorithms for learning with expert advice (Fig. 5).

### 5.1 Setup

We compare our active learning algorithm Alg. 2 (labeled OURS) with the *uncertainty* loss described in Sec. 2; UNCERTAINTY: greedy variant of our algorithm with the same measure of informativeness; ENTROPY: greedy approach that defines informativeness by the entropy of the network's softmax output; CORESET: clustering-based active learning algorithm of (Sener & Savarese, 2017; Geifman & El-Yaniv, 2017); BATCHBALD: approach based on the mutual information of points and model parameters (Kirsch et al., 2019); and UNIFORM sampling. We implemented the algorithms in Python and used the PyTorch (Paszke et al., 2017) library for deep learning.

We consider the following popular vision data sets trained on modern convolutional networks:

1. **FashionMNIST**(Xiao et al., 2017): $60,000$ grayscale images of size $28 \times 28$

2. **CIFAR10** (Krizhevsky et al., 2009): $50,000$ color images ($32 \times 32$) each belonging to one of 10 classes

3. **SVHN** (Netzer et al., 2011): $73,257$ real-world images ($32 \times 32$) of digits taken from Google Street View

4. **ImageNet** (Deng et al., 2009): more than 1.2 million images spanning 1000 classes

We used standard convolutional networks for training FashionMNIST (Xiao et al., 2017) and SVHN (Chen, 2020), and the CNN5 architecture (Nakkiran et al., 2019) and residual networks (resnets) (He et al., 2016) for our evaluations on CIFAR10 and ImageNet. The networks were trained with optimized hyper-parameters from the corresponding reference. All results were averaged over 10 trials unless otherwise stated. The full set of hyper-parameters and details of each experimental setup are provided in the supplementary material (Sec. C).

**Computation Time** Across all data sets, our algorithm took at most 3 minutes per update step. This was comparable (within a factor of 2) to that required by UNCERTAINTY and ENTROPY. However, relative to more sophisticated approaches, OURS was up to $\approx$ 12.3x faster than CORESET, due to expensive pairwise distance computations involved in clustering, and up to $\approx$ 11x faster than BATCHBALD, due to multiple ($\geq 10$) forward passes over the entire data on a network with dropout required for its Bayesian approximation (Kirsch et al., 2019); detailed timings are provided in the supplementary.

## 5.2 Evaluations on Vision Tasks

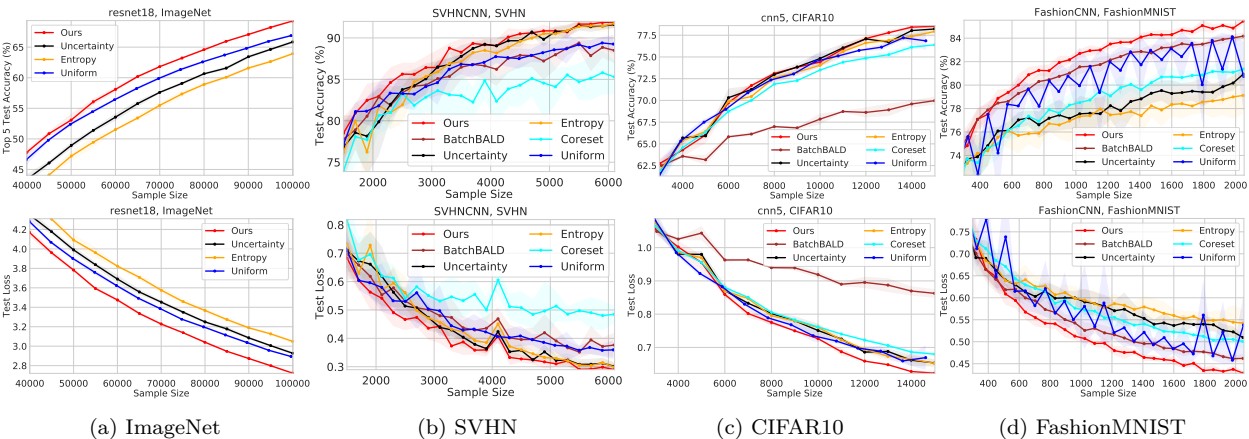

(a) ImageNet   (b) SVHN   (c) CIFAR10   (d) FashionMNIST

Figure 2: Evaluations on popular computer vision benchmarks trained on convolutional neural networks. Our algorithm consistently achieves higher performance than uniform sampling and outperforms or matches competitors on all scenarios. This is in contrast to the highly varying performance of competing methods. *Shaded regions correspond to values within one standard deviation of the mean.*

As our initial experiment, we evaluate and compare the performance of our approach on benchmark computer vision applications. Fig. 2 depicts the results of our experiments on the data sets evaluated with respect to test accuracy and test loss of the obtained network. For these experiments, we used the standard methodology (Ren et al., 2020; Gal et al., 2017) of retraining the network from scratch as the option in Alg. 1.

Note that for all data sets, our algorithm (shown in red) consistently outperforms uniform sampling, and in fact, also leads to reliable and strict improvements over existing approaches for all data sets. On ImageNet, we consistently perform better than competitors when it comes to test accuracy and loss. This difference is especially notable when we compare to greedy approaches that are outpaced by UNIFORM by up to $\approx 5\%$ test accuracy. Our results support the widespread reliability and scalability of ADAPROD$^+$, and show promise for its effectiveness on even larger models and data sets.

## 5.3 Robustness Evaluations

Next, we investigate the robustness of the considered approaches across varying data acquisition configurations evaluated on *a fixed data set*. To this end, we define a data acquisition configuration as the tuple (OPTION, $n_{start}, b, n_{end}$) where OPTION is either SCRATCH or INCR in the context of Alg. 1, $n_{start}$ is the number of initial points at the first step of the active learning iteration, $b$ is the fixed label budget per iteration, and $n_{end}$ is the number of points at which the active learning process stops. Intuitively, we expect robust active learning algorithms to be resilient to changes in the data acquisition configuration and to outperform uniform sampling in a configuration-agnostic way.

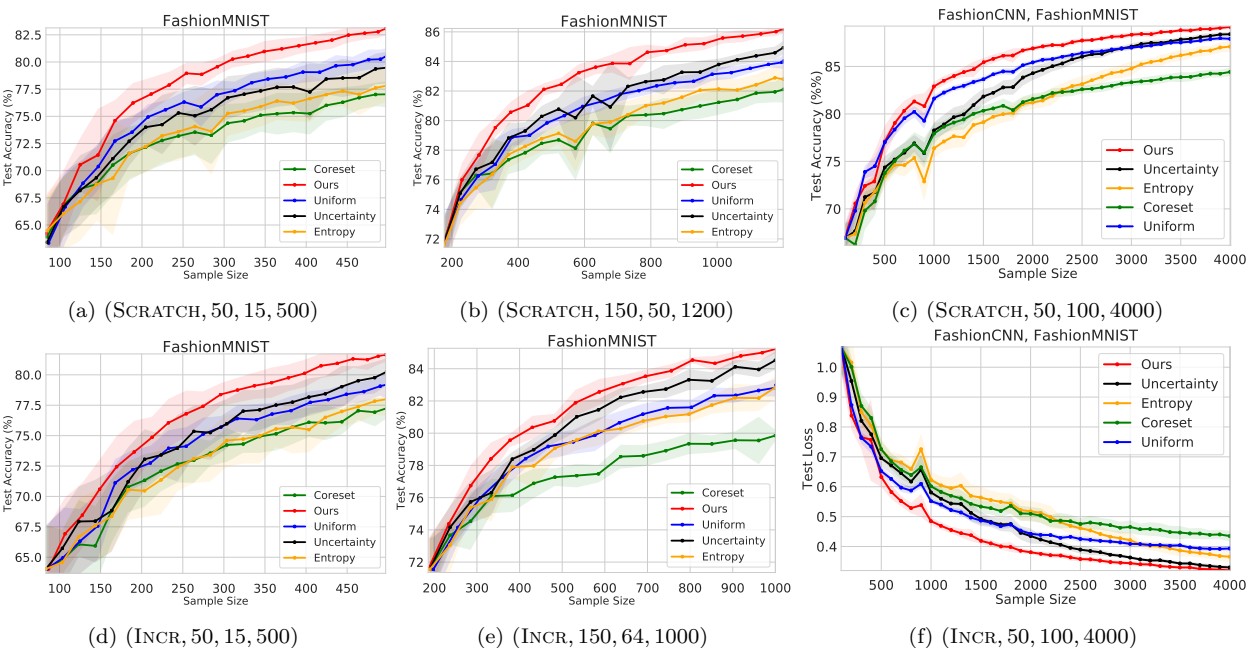

Figure 3: Our evaluations on the FashionMNIST data set with varying data acquisition configurations and INCR and SCRATCH – (OPTION, $n_{\text{start}}, b, n_{\text{end}}$). All figures except for (f) depict the test accuracy. The performance of competing methods varies greatly across configurations even when the data set is fixed.

Fig. 3 shows the results of our experiments on FashionMNIST. From the figures, we can see that our approach performs significantly better than the compared approaches in terms of both test accuracy and loss in all evaluated configurations. In fact, the compared methods' performance fluctuates wildly, supporting our premise about greedy acquisition. For instance, we can see that the uncertainty metric in Fig. 3 fares worse than naive uniform sampling in (a), but outperforms UNIFORM in settings (d) and (e); curiously, in (c), it is only better after an interesting cross-over point towards the end.

This inconsistency and sub-uniform performance is even more pronounced for the ENTROPY and CORESET algorithms that tend to perform significantly worse – up to -7% and -4% (see (a) and (e) in Fig. 3) absolute test accuracy when compared to that of our method and uniform sampling, respectively. We postulate that the poor performance of these competing approaches predominantly stems from their inherently greedy acquisition of data points in a setting with significant randomness as a result of stochastic training and data augmentation, among other elements. In contrast, our approach has provably low-regret with respect to the data acquisition objective, and we conjecture that this property translates to consistent performance across varying configurations and data sets.

## 5.4 Boosting Prior Approaches

Despite the favorable results presented in the previous subsections, a lingering question still remains: to what extent is our choice of the loss as the uncertainty metric responsible for the effectiveness of our approach? More generally, can we expect our algorithm to perform well off-the-shelf – and even lead to improvements over greedy acquisition – with other choices for the loss? To investigate, we implement three variants of our approach, OURS (UNCERTAINTY), OURS (ENTROPY), and OURS (BALD) that are instantiated with losses defined in terms of uncertainty, entropy, BALD metrics respectively, and compare to their corresponding greedy variants on SVHN and FashionMNIST. We note that the uncertainty loss corresponds to $\ell_{t,i} = \max_j f_t(x_i)_j \in [0, 1]$ and readily fits in our framework. For the ENTROPY and BALD loss, the application is only slightly more nuanced in that we have to be careful that losses are bounded in the interval $[0, 1]$. This can be done by scaling the losses appropriately, e.g., by normalizing the losses for each round to be in $[0, 1]$ or scaling using a priori knowledge, e.g., the maximum entropy is $\log(k)$ for a classification task with $k$ classes.

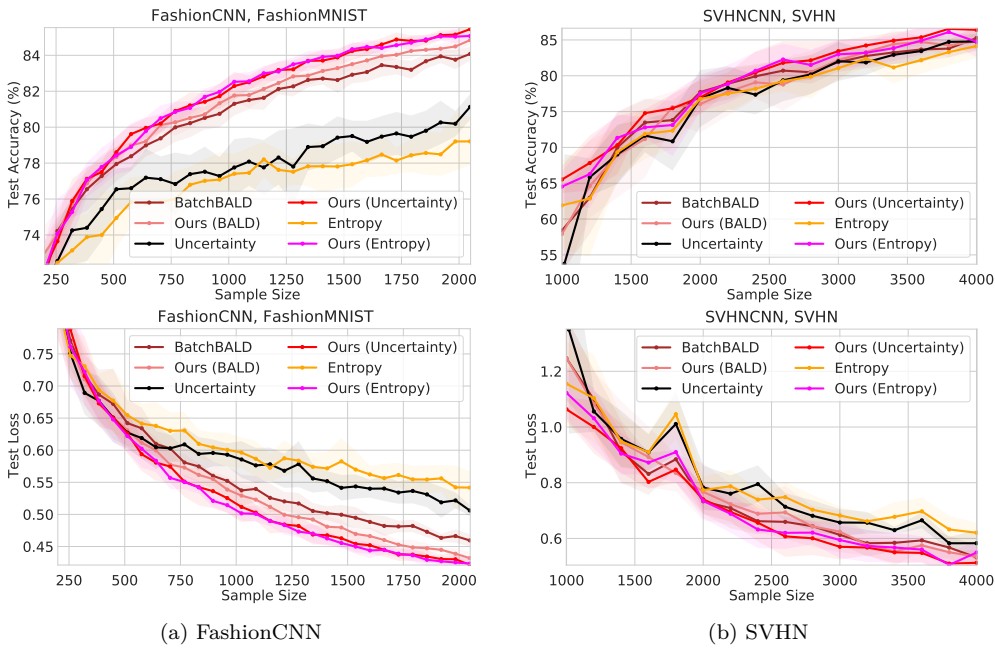

Figure 4: The performance of our algorithm when instantiated with informativeness metrics from prior work compared to that of existing greedy approaches. Using ADAPROD$^+$ off-the-shelf with the corresponding metrics took only a few lines of code and lead to strict gains in performance on all evaluated benchmark data sets.

The performance of the compared algorithms are shown in Fig. 4. Note that for all evaluated data sets and metrics, our approach fares significantly better than its greedy counterpart. In other words, applying ADAPROD$^+$ off-the-shelf with existing informativeness measures leads to strict improvements compared to their greedy variants. As seen from Fig. 4, our approach has potential to yield up to a 5% increase in test accuracy, and in all cases, achieves significantly lower test loss.

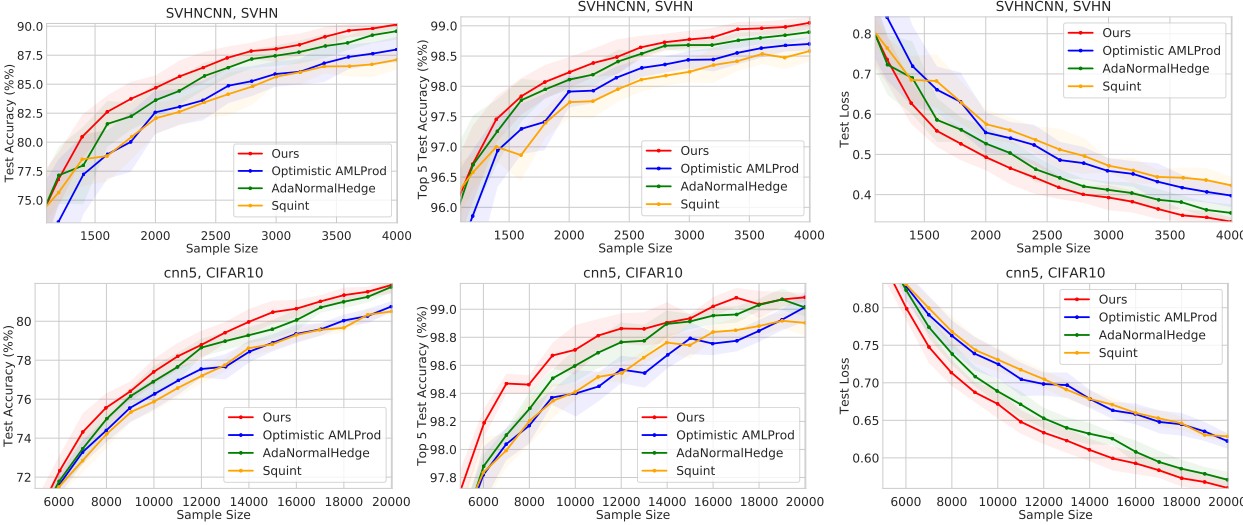

Figure 5: Comparisons with competing algorithms for learning with prediction advice on the SVHN (first row) and CIFAR10 (second row) data sets. In both scenarios, ADAPROD$^*$ outperforms the compared algorithms, and significantly improves on its predecessor, Optimistic AMLProd, on both data sets and all evaluated metrics.

### 5.5 Comparison to Existing Expert Algorithms

In this section, we consider the performance of AdaProd$^+$ relative to that of state-of-the-art algorithms for learning with prediction advice. In particular, compare our approach to Optimistic AMLProd (Wei et al., 2017), AdaNormalHedge(.TV) (Luo & Schapire, 2015), and Squint(.TV) (Koolen & Van Erven, 2015) on the SVHN and CIFAR10 data sets. Fig. 5 depicts the results of our evaluations. As the figures show, our approach outperforms the compared approaches across both data sets in terms of all of the metrics considered. AdaNormalHedge comes closest to our method in terms of performance. Notably, the improved learning rate schedule (see Sec. 4) of AdaProd$^+$ compared to that of Optimistic AMLProd enables up to 3% improvements on test error on, e.g., SVHN and 2% on CIFAR10.

## 6 Conclusion

In this paper, we introduced a low-regret active learning approach based on formulating the problem of data acquisition as that of prediction with experts. Building on our insights on the existing research gap in active learning, we introduced an efficient algorithm with performance guarantees that is tailored to achieve low regret on predictable instances while remaining resilient to adversarial ones. Our empirical evaluations on large-scale real-world data sets and architectures substantiate the reliability of our approach in outperforming naive uniform sampling and show that it leads to consistent and significant improvements over existing work. Our analysis and evaluations suggest that AdaProd$^+$ can be applied off-the-shelf with existing informativeness measures to improve upon greedy selection, and likewise can scale with future advances in uncertainty or informativeness quantification. In this regard, we hope that this work can contribute to the advancement of reliably effective active learning approaches that can one day become an ordinary part of every practitioner's toolkit, just like Adam and SGD have for stochastic optimization.

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

In this supplementary material, we provide the full proofs of our analytical results (Sec. A), implementation details of our full algorithm capable of batch sampling (Sec. B), details of experiments and additional evaluations (Sec. C), and a discussion of limitations and future work (Sec. D).

## A   Analysis

In this section, we present the full proofs and technical details of the claims made in Sec. 4. The outline of our analysis as follows. We first consider the *base* ADAPROD$^+$ algorithm (shown as Alg. 3), which is nearly the same algorithm as ADAPROD$^+$, with the exception that it is meant to be a general purpose algorithm for a setting with $K$ experts ($K$ is not necessarily equal to the number of points $n$). We show that this algorithm retains the original regret guarantees with respect to a stationary competitor of Adapt-ML-Prod.

We then consider the thought experiment where we use this standard version of our algorithm with the $K = nT$ sleeping experts reduction shown in (Wei et al., 2017; Gaillard et al., 2014) to obtain guarantees for adaptive regret. This leads us to the insight (as in (Luo & Schapire, 2015; Koolen & Van Erven, 2015)) that we do not need to keep track of the full set of $K$ experts, and can instead keep track of a much smaller (but growing) set of experts in an efficient way without compromising the theoretical guarantees.

---

**Algorithm 3** BASE ADAPROD$^+$

---

1: For all $i \in [K]$, $C_{i,0} \leftarrow 0$;   $\eta_{0,i} \leftarrow \sqrt{\log(K)/2}$; $w_{0,i} = 1$;   $\hat{r}_{1,i} = 0$;

2: **for** each round $t \in [T]$ **do**

3:    $p_{t,i} \leftarrow \eta_{t-1,i} w_{t-1,i} \exp(\eta_{t-1,i} \hat{r}_{t,i})$ for each $i \in [K]$

4:    $p_{t,i} \leftarrow p_{t,i} / \sum_{j \in [K]} p_{t,j}$ for each $i \in [K]$ {Normalize}

5:    Adversary reveals $\ell_t$ and we suffer loss $\tilde{\ell}_t = \langle \ell_t, p_t \rangle$

6:    For all $i \in [K]$, set $r_{t,i} \leftarrow \tilde{\ell}_t - \ell_{t,i}$

7:    For all $i \in [K]$, set $C_{t,i} \leftarrow C_{t-1,i} + (\hat{r}_{t,i} - r_{t,i})^2$

8:    Get prediction $\hat{r}_{t+1} \in [-1,1]^K$ for next round (see Sec. 3.2)

9:    For all $i \in [K]$, update the learning rate

$$\eta_{t,i} \leftarrow \min\left\{ \eta_{t-1,i}, \frac{2}{3(1+\hat{r}_{t+1,i})}, \sqrt{\frac{\log(K)}{C_{t,i}}} \right\}$$

10:    For all $i \in [K]$, update the weights

$$w_{t,i} \leftarrow \left( w_{t-1,i} \exp\left( \eta_{t-1,i} r_{t,i} - \eta_{t-1,i}^2 (r_{t,i} - \hat{r}_{t,i})^2 \right) \right)^{\eta_{t,i}/\eta_{t-1,i}}$$

11: **end for**

---

### A.1   Recovering Optimistic Adapt-ML-Prod Guarantees for Alg. 3

We begin by observing that Alg. 3 builds on the standard Optimistic Adapt-ML-Prod algorithm (Wei et al., 2017) by using a different initialization of the variables (Line 1) and upper bound imposed on the learning rates (as in Alg. 2, and analogously, in Line 9 of Alg. 3). Hence, the proof is has the same structure as (Wei et al., 2017; Gaillard et al., 2014), and we prove all of the relevant claims (at times, in slightly different ways) below for clarity and completeness. We proceed with our key lemma about the properties of the learning rates.

**Lemma 4** (Properties of Learning Rates). *Assume that the losses are bounded $\ell_t \in [0,1]^K$ and that the learning rates $\eta_{t,i}$ are set according to Line 9 of Alg. 3 for all $t \in [T]$ and $i \in [K]$, i.e.,*

$$\eta_{t,i} \leftarrow \min\left\{\eta_{t-1,i}, \frac{2}{3(1+\hat{r}_{t+1,i})}, \sqrt{\frac{\log(K)}{C_{t,i}}}\right\}.$$

*Then, all of the following hold for all $t \in [T]$ and $i \in [K]$:*

*1.* $\eta_{t,i}(r_{t+1,i} - \hat{r}_{t+1,i}) - \eta_{t,i}^2(r_{t+1,i} - \hat{r}_{t+1,i})^2 \leq \log\left(1 + \eta_{t,i}(r_{t+1,i} - \hat{r}_{t+1,i})\right),$

*2.* $x \leq x^{\eta_{t,i}/\eta_{t+1,i}} + 1 - \frac{\eta_{t+1,i}}{\eta_{t,i}} \quad \forall x \geq 0,$

*3.* $\frac{\eta_{t,i} - \eta_{t+1,i}}{\eta_{t,i}} \leq \log(\eta_{t,i}/\eta_{t+1,i})$ .

*Proof.* For the first claim, observe that the range of admissible values in the original Prod inequality (Cesa-Bianchi & Lugosi, 2006)

$$\forall x \geq -1/2 \quad x - x^2 \leq \log(1+x)$$

can be improved[2] to $\forall x \geq -2/3$. Now let $x = \eta_{t,i}(r_{t+1,i} - \hat{r}_{t+1,i})$, and observe that since $\ell_t \in [0,1]^K$, we have $r_{t+1,i} = \langle p_{t+1}, \ell_{t+1}\rangle - \ell_{t+1} \in [-1,1]$, and so

$$x \geq \eta_{t,i}(-1 - \hat{r}_{t+1,i}) = -\eta_{t,i}(1 + \hat{r}_{t+1,i})$$
$$\geq -2/3,$$

where in the last inequality we used the upper bound on $\eta_{t,i} \leq 2/(3(1+\hat{r}_{t+1,i}))$ which holds by definition of the learning rates.

For the second claim, recall Young's inequality[3] which states that for non-negative $a, b$, and $p \geq 1$,

$$ab \leq a^p/p + b^{p/(p-1)}(1 - 1/p).$$

For our application, we set $a = x$, $b = 1$, and $p = \eta_{t,i}/\eta_{t+1,i}$. Observe that $p$ is indeed greater than 1 since the learning rates are non-increasing over time (i.e., $\eta_{t+1,i} \leq \eta_{t,i}$ for all $t$ and $i$) by definition. Applying Young's inequality, we obtain

$$x \leq x^{\eta_{t,i}/\eta_{t+1,i}}(\eta_{t+1,i}/\eta_{t,i}) + \frac{\eta_{t,i} - \eta_{t+1,i}}{\eta_{t,i}},$$

and the claim follows by the fact that the learning rates are non-increasing.

For the final claim, observe that the derivative of $\log(x)$ is $1/x$, and so by the mean value theorem we know that there exists $c \in [\eta_{t+1,i}, \eta_{t,i}]$ such that

$$\frac{\log(\eta_{t,i}) - \log(\eta_{t+1,i})}{\eta_{t,i} - \eta_{t+1,i}} = \frac{1}{c}.$$

Rearranging and using $c \leq \max\{\eta_{t,i}, \eta_{t+1,i}\} = \eta_{t,i}$, we obtain

$$\log(\eta_{t,i}/\eta_{t+1,i}) = \frac{\eta_{t,i} - \eta_{t+1,i}}{c} \geq \frac{\eta_{t,i} - \eta_{t+1,i}}{\eta_{t,i}}.$$

$\square$

Having established our helper lemma, we now proceed to bound the regret with respect to a single expert as in (Wei et al., 2017; Gaillard et al., 2014). The main statement is given by the lemma below.

---

[2] By inspection of the root of the function $g(x) = \log(1+x) - x + x^2$ closest to $x = -1/2$, which we know exists since $g(-1/2) > 0$ while $g(-1) < 0$.

[3] This follows by taking logarithms and using the concavity of the logarithm function.

**Lemma 5** (BASE ADAPROD$^+$ Static Regret Bound). *The static regret of Alg. 3 with respect to any expert $i \in [K]$, $\sum_t r_{t,i}$, is bounded by*

$$\mathcal{O}\left(\log K + \log \log T + (\sqrt{\log K} + \log \log T)\sqrt{C_{T,i}}\right),$$

*where $C_{T,i} = \sum_{t \in [T]}(r_{t,i} - \hat{r}_{t,i})^2$.*

*Proof.* Consider $W_t = \sum_{i \in [K]} w_{t,i}$ to be the sum of *potentials* at round $t$. We will first show an upper bound on the potentials and then show that this sum is an upper bound on the regret of any expert (plus some additional terms). Combining the upper and lower bounds will lead to the statement of the lemma. To this end, we first show that the sum of potentials does not increase too much from round $t$ to $t + 1$. To do so, we apply (2) from Lemma 4 with $x = w_{t+1,i}$ to obtain for each $w_{t+1,i}$

$$w_{t+1,i} \leq w_{t+1,i}^{\eta_{t,i}/\eta_{t+1,i}} + \frac{\eta_{t,i} - \eta_{t+1,i}}{\eta_{t,i}}.$$

Now consider the first term on the right hand side above and note that

$$
\begin{aligned}
w_{t+1,i}^{\eta_{t,i}/\eta_{t+1,i}} &= w_{t,i} \exp\left(\eta_{t,i} r_{t+1,i} - \eta_{t,i}^2 (r_{t+1,i} - \hat{r}_{t+1,i})^2\right) && \text{by definition; see Line 10} \\
&= w_{t,i} \exp(\eta_{t,i}\hat{r}_{t+1,i}) \exp\left(\eta_{t,i}(r_{t+1,i} - \hat{r}_{t+1,i}) - \eta_{t,i}^2(r_{t+1,i} - \hat{r}_{t+1,i})^2\right) && \text{adding and subtracting } \eta_{t,i}\hat{r}_{t+1,i} \\
&\leq w_{t,i} \exp(\eta_{t,i}\hat{r}_{t+1,i})\left(1 + \eta_{t,i}(r_{t+1,i} - \hat{r}_{t+1,i})\right) && \text{by (1) of Lemma 4} \\
&= w_{t,i}\eta_{t,i}\exp(\eta_{t,i}\hat{r}_{t+1,i})r_{t+1,i} + w_{t,i}\exp(\eta_{t,i}\hat{r}_{t+1,i})\underbrace{(1 - \eta_{t,i}\hat{r}_{t+1,i})}_{\leq \exp(-\eta_{t,i}\hat{r}_{t+1,i})} && (1 + x \leq e^x \text{ for all real x}) \\
&\leq \underbrace{w_{t,i}\eta_{t,i}\exp(\eta_{t,i}\hat{r}_{t+1,i})}_{\propto p_{t+1,i}} r_{t+1,i} + w_{t,i}.
\end{aligned}
$$

As the brace above shows, the first part of the first expression on the right hand side is proportional to $p_{t+1,i}$ by construction (see Line 3 in Alg. 3). Recalling that $r_{t+1,i} = \langle p_{t+1}, \ell_{t+1} \rangle - \ell_{t+1,i}$, we have by dividing and multiplying by the normalization constant,

$$\sum_{i \in [K]} w_{t,i}\eta_{t,i}\exp(\eta_{t,i}\hat{r}_{t+1,i})r_{t+1,i} = \left(\sum_{i \in [K]} w_{t,i}\eta_{t,i}\exp(\eta_{t,i}\hat{r}_{t+1,i})\right)\sum_{i \in [K]} p_{t+1,i}r_{t+1,i} = 0,$$

since $\sum_{i \in [K]} p_{t+1,i}r_{t+1,i} = 0$. This shows that $\sum_{i \in [K]} w_{t+1,i}^{\eta_{t,i}/\eta_{t+1,i}} \leq \sum_{i \in [K]} w_{t,i} = W_t$.

Putting it all together and applying (3) from Lemma 4 to bound $\frac{\eta_{t,i} - \eta_{t+1,i}}{\eta_{t,i}}$, we obtain for the sum of potentials for $t \in [T]$:

$$W_{t+1} \leq W_t + \sum_{i \in [K]} \log(\eta_{t,i}/\eta_{t+1,i}).$$

A subtle issue is that for $t = 0$, we have $\eta_{0,i} = \sqrt{\log(K)/2}$ for all $i \in [n]$, which means that we cannot apply (1) of Lemma 4. So, we have to bound the change in potentials between $W_1$ and $W_0$. Fortunately, since this only occurs at the start, we can use the rough upper bound $\exp(x - x^2) = \exp(x(1 - x)) \leq \exp(1/4) \leq 1.285$, which holds for all $x \in \mathbb{R}$, to obtain for $t = 0$

$$
\begin{aligned}
w_{t+1,i}^{\eta_{t,i}/\eta_{t+1,i}} &\leq w_{t,i}\exp(\eta_{t,i}\hat{r}_{t+1,i})\exp(1/4) \\
&= w_{t,i}(1 - \eta_{t,i}\hat{r}_{t+1,i} + \eta_{t,i}\hat{r}_{t+1,i})\exp(\eta_{t,i}\hat{r}_{t+1,i})\exp(1/4) \\
&\leq w_{t,i}(\exp(-\eta_{t,i}\hat{r}_{t+1,i}) + \eta_{t,i}\hat{r}_{t+1,i})\exp(\eta_{t,i}\hat{r}_{t+1,i})\exp(1/4) \\
&= \exp(1/4)w_{t,i} + \hat{r}_{t+1,i}\underbrace{\exp(1/4)w_{t,i}\eta_{t,i}\exp(\eta_{t,i}\hat{r}_{t+1,i})}_{\propto p_{t+1,i}},
\end{aligned}
$$

where we used $1 - x \leq \exp(-x)$. Summing the last expression we obtained across all $i \in [K]$, we have for $t = 0$

$$\sum_{i \in [K]} w_{t+1,i}^{\eta_{t,i}/\eta_{t+1,i}} \leq \sum_{i \in [K]} \exp(1/4) w_{t,i},$$

where we used the fact that $\sum_{i \in [K]} \hat{r}_{t+1,i} w_{t,i} \eta_{t,i} \exp(\eta_{t,i} \hat{r}_{t+1,i}) = 0$ by definition of our predictions. Putting it all together, we obtain $W_1 \leq \exp(1/4) W_0 = \exp(1/4) K$ given that $W_0 = K$.

We can now unroll the recursion in light of the above to obtain

$$\begin{aligned}
W_T &\leq K \exp(1/4) + \sum_{t \in [T]} \sum_{i \in [K]} \log(\eta_{t,i}/\eta_{t+1,i}) \\
&= K \exp(1/4) + \sum_{i \in [K]} \sum_{t \in [T]} \log(\eta_{t,i}/\eta_{t+1,i}) \\
&= K \exp(1/4) + \sum_{i \in [K]} \log \left( \prod_{t+1 \in [T]} \eta_{t,i}/\eta_{t+1,i} \right) \\
&= K \exp(1/4) + \sum_{i \in [K]} \log(\eta_{0,i}/\eta_{T,i}) \\
&\leq K \left( \exp(1/4) + \log \left( \max_{i \in [K]} \sqrt{C_{T,i}} \right) \right) \\
&\leq K \left( \exp(1/4) + \log(4T)/2 \right).
\end{aligned}$$

Now, we establish a lower bound for $W_t$ in terms of the regret with respect to any expert $i \in [K]$. Taking the logarithm and using the fact that the potentials are always non-negative, we can show via a straightforward induction (as in (Gaillard et al., 2014)) that

$$\log(W_T) \geq \log(w_{T,i}) \geq \eta_{T,i} \sum_{t \in [T]} (r_{t,i} - \eta_{t-1,i}(r_{t,i} - \hat{r}_{t,i})^2).$$

Rearranging, and using the upper bound on $W_T$ from above, we obtain

$$\sum_{t \in [T]} r_{t,i} \leq \eta_{T,i}^{-1} \log \left( K(1 + \log(\max_{i \in [K]} \sqrt{1 + C_{T,i}}) \right) + \sum_{t \in [T]} \eta_{t-1,i}(r_{t,i} - \hat{r}_{t,i})^2. \tag{2}$$

For the first term in (2), consider the definition of $\eta_{T,i}$ and note that $\eta_{T,i} \geq \min\{1/3, \eta_{T-1,i}, \sqrt{\log(K)/(C_{T,i})}\}$ since $\hat{r}_{T+1,i} \leq 1$. Now to lower bound $\eta_{T,i}$, consider the claim that $\eta_{t,i} \geq \min\{1/3, \sqrt{\log(K)/(C_{T,i})}\}$. Note that this claim holds trivially for the base cases where $t = 0$ and $t = 1$ since the learning rates are initialized to 1 and our optimistic predictions can be at most 1. By induction, we see that if this claim holds at time step $t$, we have for time step $t + 1$

$$\begin{aligned}
\eta_{t+1,i} &\geq \min\{1/3, \eta_{t,i}, \sqrt{\log(K)/(C_{t+1,i})}\} \geq \min\{1/3, \eta_{t,i}, \sqrt{\log(K)/(C_{T,i})}\} \\
&= \min\{\eta_{t,i}, \min\{1/3, \sqrt{\log(K)/(C_{T,i})}\}\} \\
&\geq \min \left\{ \min\{1/3, \sqrt{\log(K)/(C_{T,i})}\}, \min\{1/3, \sqrt{\log(K)/(C_{T,i})}\} \right\} \\
&= \min\{1/3, \sqrt{\log(K)/(C_{T,i})}\}.
\end{aligned}$$

Hence, we obtain $\eta_{T,i} \geq \min\{1/3, C_{T,i}\}$, and this implies that (by the same reasoning as in (Gaillard et al., 2014)) that

$$\eta_{T,i}^{-1} \log \left( K(1 + \log(\max_{i \in [K]} \sqrt{C_{T,i}}) \right) \leq \mathcal{O} \left( (\sqrt{\log K} + \log \log T) \sqrt{C_{T,i}} + \log K \right).$$

Now to bound the second term in (2), $\sum_{t \in [T]} \eta_{t-1,i}(r_{t,i} - \hat{r}_{t,i})^2$, we deviate from the analysis in (Wei et al., 2017) in order to show that the improved learning schedule without the dampening term in the denominator suffices. To this end, we first upper bound $\eta_{t-1,i}$ as follows

$$\eta_{t-1,i} \leq \min\left\{\eta_{0,i}, \frac{2}{3(1+\hat{r}_{t,i})}, \sqrt{\frac{\log(K)}{C_{t-1,i}}}\right\}$$

$$\leq \min\left\{\eta_{0,i}, \sqrt{\frac{\log(K)}{C_{t-1,i}}}\right\}$$

$$= \min\left\{\eta_{0,i}, \eta_{0,i}\sqrt{\frac{2}{C_{t-1,i}}}\right\}$$

where first inequality follows from the fact that the learning rates are monotonically decreasing, the second inequality from the definition of min, the last equality by definition $\eta_{0,i} = \sqrt{\log(K)/2}$.

By the fact that the minimum of two positive numbers is less than its harmonic mean[4], we have

$$\eta_{t-1,i} \leq \frac{2\sqrt{2}\eta_{0,i}}{\sqrt{2} + \sqrt{C_{t-1,i}}},$$

and so

$$(r_{t,i} - \hat{r}_{t,i})^2 \eta_{t-1,i} \leq c_{t,i} \frac{2\sqrt{2}\eta_{0,i}}{\sqrt{2} + \sqrt{C_{t-1,i}}}$$

$$= 8\sqrt{2}\eta_{0,i} \frac{(c_{t,i}/4)}{\sqrt{2} + 2\sqrt{C_{t-1,i}/4}}$$

$$\leq 4\sqrt{2}\eta_{0,i} \frac{(c_{t,i}/4)}{\sqrt{1/2 + C_{t-1,i}/4}},$$

where we used the subadditivity of the square root function in the last step.

Summing over all $t \in [T]$ and applying Lemma 14 of (Gaillard et al., 2014) on the scaled variables $c_{t,i}/4 \in [0,1]$, we obtain

$$\sum_{t \in [T]} \eta_{t-1,i}(r_{t,i} - \hat{r}_{t,i})^2 \leq 4\sqrt{2}\eta_{0,i} \sum_{t \in [T]} \sqrt{C_{T,i}}$$

$$= 4\sqrt{C_{T,i}\log K},$$

where in the last equality we used the definition of $\eta_{i,0}$ and $C_{T,i} = \sum_{t \in [T]}(r_{t,i} - \hat{r}_{t,i})^2$ as before, and this completes the proof.

$\square$

## A.2  Adaptive Regret

We now turn to establishing adaptive regret bounds via the sleeping experts reduction as in (Wei et al., 2017; Luo & Schapire, 2015) using the reduction of (Gaillard et al., 2014). The overarching goal is to establish an adaptive bound for the regret of *every* time interval $[t_1, t_2], t_1, t_2 \in [T]$, which is a generalization of the static regret which corresponds to the regret over the interval $[1, T]$. To do so, in the setting of $n$ experts as in the main document, the main idea is to run the base algorithm (Alg. 3) on $K = nT$ *sleeping* experts instead[5]. These experts will be indexed by $(t, i)$ with $t \in [T]$ and $i \in [n]$. Moreover, at time step $t$, each expert $(s, i)$ is

---

[4]$\min\{a, b\} = \min\{a^{-1}, b^{-1}\}^{-1} \leq (\frac{1}{2}(a^{-1} + b^{-1}))^{-1} = 2/(1/a + 1/b)$
[5]Note that this notion of sleeping experts is the same as the one we used for dealing with constructing a distribution over only the unlabeled data points remaining.

defined to be awake if $s \leq t$, $i \in [n]$ **and** $\mathcal{I}_{t,i} = 1$ (the point has not yet been sampled, see Sec. 2), and the remaining experts will be considered sleeping. This will generate a probability distribution $\bar{p}_{t,(s,i)}$ over the awake experts. Using this distribution, at round $t$ we play

$$p_{t,i} = \mathcal{I}_{t,i} \sum_{s \in [t]} \bar{p}_{t,(s,i)} / Z_t,$$

where $Z_t = \sum_{j \in [K]} \mathcal{I}_{t,j} \sum_{s' \in [t]} \bar{p}_{t,(s',j)}$.

The main idea is to construct losses to give to the base algorithm so that that at any point $t \in [T]$, each expert $(s,i)$ suffers the interval regret from $s$ to $t$ (which is defined to be 0 if $s > t$), i.e., $\sum_{\tau=1}^{t} r_{\tau,(s,i)} = \sum_{\tau=s}^{t} r_{\tau,i}$. To do so, we build on the reduction of (Wei et al., 2017) to keep track of both the sleeping experts from the sense of achieving adaptive regret and also the traditional sleeping experts regret with respect to only those points that are not yet labeled (as in Sec. 2). The idea is to apply the base algorithm (Alg. 3) with the modified loss vectors $\bar{\ell}_{t,(s,i)}$ for expert $(s,i)$ as the original loss if the expert is awake, i.e., $\bar{\ell}_{t,(s,i)} = \ell_{t,i}$ **if** $s \leq t$ (original reduction in (Wei et al., 2017)) **and** $\mathcal{I}_{t,i} = 1$ (the point has not yet been sampled), and $\bar{\ell}_{t,(s,i)} = \langle p_{t,i}, \ell_t \rangle$ otherwise. The prediction vector is defined similarly: $\bar{r}_{t,(s,i)} = \hat{r}_{t,i}$ if $s \leq t$, and 0 otherwise.

Note that this construction implies that the regret of the base algorithm with respect to the modified losses and predictions, i.e., $\bar{r}_{\tau,(s,i)} = \langle \bar{p}_{\tau,(s,i)}, \bar{\ell}_{\tau,(s,i)} \rangle$ is equivalent to $r_{\tau,i}$ for rounds $\tau > s$ where the expert is awake, and 0 otherwise. Thus,

$$\sum_{\tau \in [t]} \bar{r}_{\tau,(s,i)} = \sum_{\tau=s}^{t} r_{\tau,i},$$

which means that the regret of expert $(s,i)$ with respect to the base algorithm is precisely regret of the interval $[s,t]$. Applying Lemma 5 to this reduction above (with $K = nT$) immediately recovers the adaptive regret guarantee of Optimisic Adapt-ML-Prod.

**Lemma 6** (Adaptive Regret of BASE ADAPROD$^+$)**.** *For any $t_1 \leq t_2$ and $i \in [n]$, invoking Alg. 3 with the sleeping experts reduction described above ensures that*

$$\sum_{t=t_1}^{t_2} r_{t,i} \leq \hat{\mathcal{O}}\left(\log(K) + \sqrt{C_{t_2,(t_1,i)} \log(K)}\right),$$

*where $C_{t_2,(t_1,i)} = \sum_{t=t_1}^{t_2} (r_{t,i} - \hat{r}_{t,i})^2$ and $\hat{\mathcal{O}}$ suppresses $\log T$ factors.*

### A.3 AdaProd$^+$ and Proof of Lemma 1

To put it all together, we relax to requirement of having to update and keep track of $K = NT$ experts and having to know $T$. To do so, observe that $\log(K) \leq \log(nT) \leq 2\log(n)$ since $T \leq n / \min_{t \in [T]} b_t \leq n$, where $b_t \geq 1$ is the number of new points to label at active learning iteration $t$. This removes the requirement of having to know $T$ or the future batch sizes beforehand, meaning that we can set the numerator of $\eta_{t,(s,i)}$ to be $\sqrt{2\log(n)}$ instead of $\sqrt{\log(K)}$ (as in 2 in Sec. 3). Next, observe that in the sleeping experts reduction above, we have

$$p_{t,i} = \mathcal{I}_{t,i} \sum_{s \in [t]} \bar{p}_{t,(s,i)} / Z_t,$$

where $Z_t = \sum_{j \in [K]} \mathcal{I}_{t,j} \sum_{s' \in [t]} \bar{p}_{t,(s',j)}$. But for $s \leq t$ and $j \in [n]$ satisfying $\mathcal{I}_{t,j} = 1$, by definition of $\bar{p}_{t,(s,j)}$ and the fact that expert $(s,j)$ is awake, we have $\bar{p}_{t,(s,j)} \propto \eta_{t-1,(s,j)} w_{t-1,(s,j)} \exp(\eta_{t-1,(s,j)} \hat{r}_{t,j})$, and so the normalization constant cancels from the numerator (from $\bar{p}_{t,(s,j)}$) and the denominator (from the $\bar{p}_{t,(s',j)}$ in $Z_t = \sum_{j \in [K]} \mathcal{I}_{t,j} \sum_{s' \in [t]} \bar{p}_{t,(s',j)}$), leaving us with

$$p_{t,i} = \sum_{s \in [t]} \frac{\eta_{t-1,(s',j)} w_{t-1,(s',j)} \exp(\eta_{t-1,(s',j)} \hat{r}_{t,j})}{\gamma_t},$$

where $\gamma_t = \sum_{j \in [K]} \sum_{s' \in [t]} \eta_{t-1,(s',j)} w_{t-1,(s',j)} \exp(\eta_{t-1,(s',j)} \hat{r}_{t,j})$. Note that this corresponds precisely to the probability distribution played by ADAPROD$^+$. Further, since ADAPROD$^+$ does not explicitly keep track

of the experts that are asleep, and only updates the potentials $W_{t,(s,i)}$ of those experts that are awake, AdaProd$^+$ mimics the updates of the reduction described above[6] involving passing of the modified losses to the base algorithm. Thus, we can conclude that AdaProd$^+$ leads to the same updates and generated probability distributions as the base algorithm for adaptive regret. This discussion immediately leads to the following lemma for the adaptive regret of our algorithm, very similar to the one established above except for $\log n$ replacing $\log T$ terms.

**Lemma 1** (Adaptive Regret of AdaProd$^+$). *For any $t_1 \leq t_2$ and $i \in [n]$, Alg. 2 ensures that*

$$\sum_{t=t_1}^{t_2} r_{t,i} \leq \mathcal{O}\left(\log n + \log\log n + (\sqrt{\log n} + \log\log n)\sqrt{C_{t_2,(t_1,i)}}\right),$$

*where $C_{t_2,(t_1,i)} = \sum_{t=t_1}^{t_2}(r_{t,i} - \hat{r}_{t,i})^2$ and $r_{t,i} = (\langle \ell_t, p_t \rangle - \ell_{t,i})\mathcal{I}_{t,i}$ is the instantaneous regret of $i \in [n]$ at time $t$ and $\hat{r}_{t,i} = (\langle \hat{\ell}_t, p_t \rangle - \hat{\ell}_{t,i})\mathcal{I}_{t,i}$ is the predicted instantenous regret as a function of the optimistic loss vector $\hat{\ell}_t$ prediction.*

### A.4 Proof of Theorem 2

**Theorem 2** (Dynamic Regret). *AdaProd$^+$ takes at most $\tilde{\mathcal{O}}(tn_t)$ [7] time for the $t^{th}$ update and for batch size $b = 1$ for all $t \in [T]$, guarantees that over $T$ steps,*

$$\max_{(i_t)_{t\in[T]}} \mathbb{E}\left[\mathcal{R}(i_{1:T})\right] \leq \hat{\mathcal{O}}\left(\sqrt[3]{\mathbb{E}\left[\mathcal{V}_T\right]\mathcal{D}_T T \log n} + \sqrt{\mathcal{D}_T T \log n}\right),$$

*where $\hat{\mathcal{O}}(\cdot)$ suppresses $\log T$ factors.*

*Proof.* Fix $(i_t^*)_{t\in[T]}$ be an arbitrary competitor sequence. First observe that the expected regret over all the randomness as defined Sec. 2 can be written as

$$\mathbb{E}\left[\mathcal{R}(i_{1:T}^*)\right] = \mathbb{E}\left[\sum_{t\in[T]} \mathbb{E}_{\xi_{t-1}}\left[r_{t,i_t^*}\right]\right]$$

$$= \mathbb{E}_{\mathcal{S}_{1:T-1},\xi_{0:T-1}}\left[\sum_{t\in[T]} r_{t,i_t^*}\right]$$

which follows by linearity of expectation and the fact that $p_t$ is independent of $\xi_{t-1}$ and the noises $\xi_{1:T}$ are independent random variables.

Similar to the approach of (Wei et al., 2017; Besbes et al., 2014), consider partitioning the time horizon $T$ into $N = \lceil T/B \rceil$ contiguous time blocks $\mathcal{T}_1, \ldots, \mathcal{T}_N$ of length $B$, with the possible exception of $\mathcal{T}_N$ which has length at most $B$. For any time block $\mathcal{T}$, let $i_\mathcal{T}$ denote the best sample with respect to the expected regret, i.e.,

$$i_\mathcal{T} = \operatorname*{argmax}_{i\in[n]} \mathbb{E}\left[\sum_{t\in\mathcal{T}} r_{t,i}\right],$$

where the expectation is with respect to all sources of randomness. Continuing from above, Note that the expected dynamic regret in Sec. 2 can be decomposed as follows:

$$\mathbb{E}\left[\mathcal{R}(i_{1:T}^*)\right] = \mathbb{E}\left[\sum_{b=1}^N \sum_{t\in\mathcal{T}_b} r_{t,i_t^*} - r_{t,i_{\mathcal{T}_b}}\right] + \mathbb{E}\left[\sum_{b=1}^N \sum_{t\in\mathcal{T}_b} r_{t,\mathcal{T}_b}\right]$$

---

[6]The only minor change is in the constant in the learning rate schedule of AdaProd$^+$ which has a $\sqrt{2\log(n)}$ term instead of $\sqrt{\log(nT)} \leq \sqrt{2\log(n)}$. This only affects the regret bounds by at most a factor of $\sqrt{2}$, and the reduction remains valid – it would be analogous to running Alg. 3 on a set of $n^2 \geq nT$ experts instead.

[7]We use $\tilde{\mathcal{O}}(\cdot)$ to suppress $\log T$ and $\log n$ factors.

$$= \underbrace{\sum_{b=1}^{N} \sum_{t \in \mathcal{T}_b} \mathbb{E}\left[r_{t,i_t^*} - r_{t,i_{\mathcal{T}_b}}\right]}_{(A)} + \underbrace{\sum_{b=1}^{N} \mathbb{E}\left[\sum_{t \in \mathcal{T}_b} r_{t,i_{\mathcal{T}_b}}\right]}_{(B)},$$

where the last equality follows by linearity of expectation.

To deal with the first term, consider an arbitrary time block $\mathcal{T}$ and define the drift in expected regret

$$\mathcal{D}_{\mathcal{T}} = \sum_{t \in [\mathcal{T}]} \|\mathbb{E}\left[r_t\right] - \mathbb{E}\left[r_{t-1}\right]\|_{\infty}.$$

For each $t \in \mathcal{T}$ and $i_t^* \in [n]$ we know that there must exist $t_0 \in \mathcal{T}$ such that $\mathbb{E}\left[r_{t_0,i_{\mathcal{T}}}\right] \geq \mathbb{E}\left[r_{t_0,i_t^*}\right]$. To see this, note that the negation of this statement would imply $\sum_{t \in \mathcal{T}} \mathbb{E}\left[r_{t,i_{\mathcal{T}}}\right] \leq \sum_{t \in \mathcal{T}} \mathbb{E}\left[r_{t,i_t^*}\right]$ which contradicts the optimality of $i_{\mathcal{T}}$. For any $t \in \mathcal{T}$, we have

$$\mathbb{E}\left[r_{t,i_t^*}\right] \leq \mathbb{E}\left[r_{t_0,i_t^*}\right] + \mathcal{D}_{\mathcal{T}} \leq \mathbb{E}\left[r_{t_0,i_{\mathcal{T}}}\right] + \mathcal{D}_{\mathcal{T}} \leq \mathbb{E}\left[r_{t,i_{\mathcal{T}}}\right] + 2\mathcal{D}_{\mathcal{T}},$$

where the first and third inequalities are by the definition of $\mathcal{D}_{\mathcal{T}}$ and the second by the argument above over $t_0$. This implies that

$$\sum_{t \in \mathcal{T}} \mathbb{E}\left[r_{t,i_t^*} - r_{t,i_{\mathcal{T}_b}}\right] \leq 2B\mathcal{D}_{\mathcal{T}}.$$

Summing over $N$ blocks, we obtain that

$$(A) \leq 2B\mathcal{D}_T = 2(T/N)\mathcal{D}_T,$$

where $\mathcal{D}_T = \sum_{t \in [T]} \|\mathbb{E}\left[r_t\right] - \mathbb{E}\left[r_{t-1}\right]\|_{\infty}$.

For the second term, we apply the adaptive regret bound of Lemma 1 to each block $\mathcal{T}_b$ ranging from time $t_b^1$ to $t_b^2$ to obtain

$$\mathbb{E}\left[\sum_{t \in \mathcal{T}_b} r_{t,i_{\mathcal{T}_b}}\right] \leq \hat{\mathcal{O}}\left(\mathbb{E}\left[\sqrt{C_{t_b^2,(t_b^1,i_{\mathcal{T}_b})}}\right] + \log n\right).$$

Summing over all $N$ blocks we have

$$(B) = \sum_{b=1}^{N} \mathbb{E}\left[\sum_{t \in \mathcal{T}_b} r_{t,i_{\mathcal{T}_b}}\right] \leq \hat{\mathcal{O}}\left(\mathbb{E}\left[\sum_{b=1}^{N} \sqrt{C_{t_b^2,(t_b^1,i_{\mathcal{T}_b})} \log n}\right] + N \log n\right)$$

$$\leq \hat{\mathcal{O}}\left(\mathbb{E}\left[\sum_{b=1}^{N} \sqrt{\sum_{t \in [\mathcal{T}_b]} \|r_t - \hat{r}_t\|_{\infty}^2 \log n}\right] + N \log n\right)$$

$$\leq \hat{\mathcal{O}}\left(\mathbb{E}\left[\sqrt{N\mathcal{V}_T \log n}\right] + N \log n\right),$$

where the second to last inequality follows by the definition of $C_{t_b^2,(t_b^1,i_{\mathcal{T}_b})}$ and the last inequality is by Cauchy-Schwarz.

Putting both bounds together, we have that

$$\mathbb{E}\left[\mathcal{R}(i_{1:T}^*)\right] = \min_{N \in [T]} \hat{\mathcal{O}}\left(\mathbb{E}\left[\sqrt{N\mathcal{V}_T \log n} + N \log n + (T/N)\mathcal{D}_T\right]\right).$$

All that remains is to optimize the bound with respect to the number of epochs $N$. If $\mathcal{V}_T^2 \leq T\mathcal{D}_T \log n$, we can pick $N = \sqrt{TD/\log n}$ to obtain a bound of $\hat{\mathcal{O}}(\sqrt{T\mathcal{D}_T \log n})$. On the other hand, if $\mathcal{V}_T^2 > T\mathcal{D}_T \log n$ we can let $N = \sqrt[3]{T^2 \mathcal{D}_T^2/(\mathcal{V}_T \log n)}$ to obtain the bound $\hat{\mathcal{O}}(\sqrt[3]{T\mathcal{D}_T \mathcal{V}_T \log n})$. Hence, in either case we have the upper bound

$$\hat{\mathcal{O}}\left(\mathbb{E}\left[\sqrt[3]{\mathcal{V}_T \mathcal{D}_T T \log n} + \sqrt{\mathcal{D}_T T \log n}\right]\right) \leq \hat{\mathcal{O}}\left(\sqrt[3]{\mathbb{E}\left[\mathcal{V}_T\right] \mathcal{D}_T T \log n} + \sqrt{\mathcal{D}_T T \log n}\right)$$

where the inequality is by Jensen's.

$\square$

### A.5 Proof of Theorem 3

**Theorem 3** (Dynamic Regret). ADAPROD$^+$ *with batch sampling of $b$ points guarantees that over $T$ steps,*

$$\max_{(\mathcal{S}_t)_{t \in [T]}:|\mathcal{S}_t|=b \,\forall t} \mathbb{E}\left[\mathcal{R}(\mathcal{S}_{1:T})\right] \leq \hat{\mathcal{O}}\left(b\sqrt[3]{\mathbb{E}\left[\mathcal{V}_T\right]\mathcal{D}_T T \log n} + b\sqrt{\mathcal{D}_T T \log n}\right).$$

*Proof.* For any fixed sequence $(\mathcal{S}_1^*, \ldots, \mathcal{S}_T^*)$, we have

$$
\begin{aligned}
\mathbb{E}\left[\mathcal{R}(\mathcal{S}_{1:T}^*)\right] &= \mathbb{E}\sum_{t=1}^{T}\sum_{j=1}^{b}\mathbb{E}_{\xi_{t-1}}[r_{t,\mathcal{S}_{tj}^*}] \\
&= \mathbb{E}\sum_{j=1}^{b}\sum_{t=1}^{T}\mathbb{E}_{\xi_{t-1}}[r_{t,\mathcal{S}_{tj}^*}] \\
&= \sum_{j=1}^{b}\mathbb{E}\underbrace{\left[\sum_{t=1}^{T}(\langle p_t, \ell_t(\xi_{t-1})\rangle - \ell_{t,i}(\xi_{t-1}))\mathcal{I}_{t,i}\right]}_{\text{per point regret from Theorem 2}} \\
&\leq b\,\hat{\mathcal{O}}\left(\sqrt[3]{\mathbb{E}\left[\mathcal{V}_T\right]\mathcal{D}_T T \log n} + \sqrt{\mathcal{D}_T T \log n}\right)
\end{aligned}
$$

where we used the fact that $\rho_t = bp_t$ and the definition of $r_t$ from Sec. 4.2. $\qquad\square$

## B  Implementation Details and Batch Sampling

To sample $b$ points according to a probability distribution $p$ with $\sum_i p_i = b$, we use use the DEPROUND algorithm (Uchiya et al., 2010) shown as Alg. 4, which takes $\mathcal{O}(n)$ time.

---

**Algorithm 4** DEPROUND

---

**Inputs**: Probabilities $p \in [0,1]^n$ such that $\sum_i p_i = b$
**Output:** set of indices $\mathcal{C} \subset [n]$ of size $b$

1: **while** $\exists i \in [n]$ such that $0 < p_i < 1$ **do**
2:   Pick $i, j \in [n]$ satisfying $i \neq j$, $0 < p_i < 1$, and $0 < p_j < 1$
3:   Set $\alpha = \min(1 - p_i, p_j)$ and $\beta = \min(p_i, 1 - p_j)$
4:   Update $p_i$ and $p_j$
$$(p_i, p_j) = \begin{cases} (p_i + \alpha, p_j - \alpha) & \text{with probability } \frac{\beta}{\alpha+\beta}, \\ (p_i - \beta, p_j + \beta) & \text{with probability } 1 - \frac{\beta}{\alpha+\beta}. \end{cases}$$
5: **end while**
6: $\mathcal{C} \leftarrow \{i \in [n] : p_i = 1\}$
   **return** $\mathcal{C}$

---

In all of our empirical evaluations, the original probabilities generated by ADAPROD$^+$ were already less than $1/b$, so the capping procedure did not get invoked. We conjecture that this may be a natural consequence of the active learning setting, where we are attempting to incrementally build up a small set of labeled data among a very large pool of unlabeled ones, i.e., $b \ll n$. This description also aligns with the relatively small batch sizes widely used in active learning literature as benchmarks (Gissin & Shalev-Shwartz, 2019; Ash et al., 2019; Ren et al., 2020; Sener & Savarese, 2017; Muthakana, 2019).

The focus of our work is not on the full extension of Adapt-ML-Prod (Wei et al., 2017) to the batch setting, however, we summarize some of our ongoing and future work here for the interested reader. If we assume that the probabilities generated by ADAPROD$^+$ satisfy $p_{t,i} \leq 1/b$, which is a mild assumption in the active learning setting as evidenced by our evaluations, we establish the bound as in Sec. 4.2 for the regret defined

with respect to sampling a batch of $b$ points at each time step. In future work, we plan to relax the assumption $p_{t,i} \leq 1/b$ by building on techniques from prior work, such as by exploiting the inequalities associated with the Information (KL divergence) Projection as in (Warmuth & Kuzmin, 2008) or capping the weight potential $w_{i,t}$ as in (Uchiya et al., 2010) as soon as weights get too large (rather than modifying the probabilities).

## C  Experimental Setup & Additional Evaluations

In this section we (i) describe the experimental setup and detail hyper-parameters used for our experiments and (ii) provide additional evaluations and comparisons to supplement the results presented in the manuscript. The full code is included in the supplementary folder[8].

### C.1  Setup

|  | FashionCNN | SVHNCNN | Resnet18 | CNN5 (width=128) |
|---|---|---|---|---|
| loss | cross-entropy | cross-entropy | cross-entropy | cross-entropy |
| optimizer | Adam | Adam | SGD | Adam |
| epochs | 60 | 60 | 80 | 60 |
| epochs incremental | 15 | 15 | N/A | 15 |
| batch size | 128 | 128 | 256 | 128 |
| learning rate (lr) | 0.001 | 0.001 | 0.1 | 0.001 |
| lr decay | 0.1@(50) | 0.1@(50) | 0.1@(30, 60) | 0.1@(50) |
| lr decay incremental | 0.1@(10) | 0.1@(10) | N/A | 0.1@(10) |
| momentum | N/A | N/A | 0.9 | N/A |
| Nesterov | N/A | N/A | No | N/A |
| weight decay | 0 | 0 | 1.0e-4 | 0 |

Table 1: We report the hyperparameters used during training the convolutional architectures listed above corresponding to our evaluations on FashionMNIST, SVHN, CIFAR10, and ImageNet. except for the ones indicated in the lower part of the table. The notation $\gamma@(n_1, n_2, \ldots)$ denotes the learning rate schedule where the learning rate is multiplied by the factor $\gamma$ at epochs $n_1, n_2, \ldots$ (this corresponds to MultiStepLR in PyTorch).

Table 1 depicts the hyperparameters used for training the network architectures used in our experiments. Given an active learning configuration (OPTION, $n_{\text{start}}, b, n_{\text{end}}$), these parameters describe the training process for each choice of OPTION as follows: **Incremental** : we start the active learning process by acquiring and labeling $n_{\text{start}}$ points chosen uniformly at random from the $n$ unlabeled data points, and we train with the corresponding number of epochs and learning rate schedule listed in Table 1 under rows *epochs* and *lr decay*, respectively, to obtain $\theta_1$. We then proceed as in Alg. 1 to iteratively acquire $b$ new labeled points based on the ACQUIRE function and *incrementally train* a model starting from the model from the previous iteration, $\theta_{t-1}$. This training is done with respect to the number of corresponding epochs and learning rate schedule shown in Table 1 under *epochs incremental* and *lr decay incremental*, respectively. **Scratch** : the only difference relative to the INCREMENTAL setting is that rather than training the model starting from $\theta_{t-1}$, we train a model from a randomly initialized network at each active learning iteration with respect to the training parameters under *epochs* and *lr decay* in Table 1.

**Architectures**  We used the following convolutional networks on the specified data sets.

1. *FashionCNN (Pankajj, 2018)* (for FashionMNIST): a network with 2 convolutional layers with batch normalization and max pooling, 3 fully connected layers, and one dropout layer with $p = 0.25$ in (Pankajj, 2018). This architecture achieves over 93% accuracy when trained with the whole data set.

2. *SVHNCNN (Chen, 2020)* (for SVHN): a small scale convolutional model very similar to FashionCNN except there is no dropout layer.

3. *Resnet18 (He et al., 2016)* (for ImageNet): an 18 layer residual network with batch normalization.

4. *CNN5 (Nakkiran et al., 2019)* (for CIFAR10): a 5-layer convolutional neural network with 4 convolutional layers with batch normalization. We used the width=128 setting in the context of (Nakkiran et al., 2019).

**Settings for experiments in Sec. 5**  Prior to presenting additional results and evaluations in the next subsections, we specify the experiment configurations used for the experiments shown in the main

---

[8]Our codebase builds on the publicly available codebase of (Liebenwein, 2021; Liebenwein et al., 2019; Baykal et al., 2018).

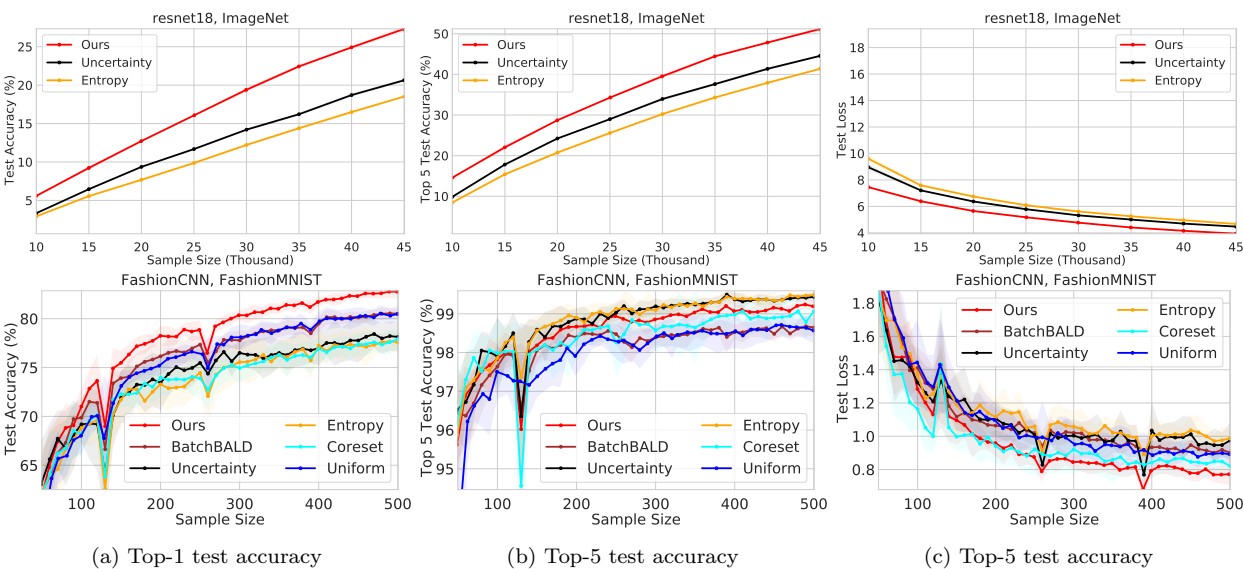

(a) Top-1 test accuracy      (b) Top-5 test accuracy      (c) Top-5 test accuracy

Figure 6: Results for the data-starved configuration ($\textsc{Scratch}, 5k, 5k, 45k$) on ImageNet (first row) and ($\textsc{Scratch}, 50, 10, 500$) on FashionMNIST (second row). Shown from left to right are the results with respect to test accuracy, top-5 test accuracy, and test loss. Shaded region corresponds to values within one standard deviation of the mean.

document (Sec. 5). For the corresponding experiments in Fig. 2, we evaluated on the configuration ($\textsc{Scratch}, 10k, 20k, 110k$) for ImageNet, ($\textsc{Scratch}, 500, 200, 4000$) for SVHN, ($\textsc{Scratch}, 3k, 1k, 15k$) for CIFAR10, and ($\textsc{Scratch}, 100, 300, 3000$) for FashionMNIST. For the evaluations in Fig. 4, we used ($\textsc{Scratch}, 128, 96, 200$) and ($\textsc{Scratch}, 128, 64, 2000$) for FashionMNIST and SVHN, respectively. The models were trained with standard data normalization with respect to the mean and standard deviation of the entire training set. For ImageNet, we used random cropping to $224 \times 224$ and random horizontal flips for data augmentation; for the remaining data sets, we used random cropping to $32 \times 32$ ($28 \times 28$ for FashionMNIST) with 4 pixels of padding and random horizontal flips.

All presented results were averaged over 10 trials with the exception of those for ImageNet[9], where we averaged over 3 trials due to the observed low variance in our results. We used the uncertainty loss metric as defined in Sec. 2 for all of the experiments presented in this work – with the exception of results related to boosting prior approaches (Fig. 4). The initial set of points and the sequence of random network initializations (one per sample size for the $\textsc{Scratch}$ option) were fixed across all algorithms to ensure fairness.

## C.2   Setting for Experiments in Sec. 5.5

In this subsection, we describe the setting for the evaluations in Sec. 5.5, where we compared the performance of $\textsc{AdaProd}^{+}$ to modern algorithms for learning with prediction advice. Since our approach is intended to compete with time-varying competitors (see Sec. A), we compare it to existing methods that ensure low regret with respect to time-varying competitors (via adaptive regret). In particular, we compare our approach to the following algorithms:

1. **Optimistic AMLProd** (Wei et al., 2017): we implement the (stronger) variant of Optimistic Adapt-ML-Prod that ensures dynamic regret (outlined at the end of Sec. 3.3 in (Wei et al., 2017)). This algorithm uses the sleeping experts reduction of (Gaillard et al., 2014) and consequently, requires initially creating $\tilde{n} = nT$ sleeping experts and updating them with similar updates as in our algorithm (except the cost of the $t^{\text{th}}$ update is $\tilde{\mathcal{O}}(nT)$ rather than $\tilde{\mathcal{O}}(N_t t)$ as in ours). Besides the computational costs, we emphasize that the only true functional difference between our algorithm and Optimistic AMLProd lies in the

---

[9]We were not able to run Coreset or BatchBALD on ImageNet due to resource constraints and the high computation requirements of these algorithms.

thresholding of the learning rates (Line 10 in Alg. 2). In our approach, we impose the upper bound $\min\{\eta_{t-1,i}, 2/(3(1+\hat{r}_{t+1,i}))\}$ for $\eta_{t,i}$ for any $t \in [T]$, whereas (Wei et al., 2017) imposes the (smaller) bound of $1/4$.

2. **AdaNormalHedge(.TV)** (Luo & Schapire, 2015): we implement the time-varying version of AdaNormalHedge, AdaNormalHedge.TV as described in Sec. 5.1 of (Luo & Schapire, 2015). The only slight modification we make in our setting where we already have a sleeping experts problem is to incorporate the indicator $\mathcal{I}_{t,i}$ in our predictions (as suggested by (Luo & Schapire, 2015) in their sleeping experts variant). In other words, we predict[10] $p_{t,i} \propto \mathcal{I}_{t,i} \sum_{\tau=1}^{t} \frac{1}{\tau^2} w(R_{[\tau,t-1],i}, C_{[\tau,t-1]})$ rather than the original $p_{t,i} \propto \sum_{\tau=1}^{t} \frac{1}{\tau^2} w(R_{[\tau,t-1],i}, C_{[\tau,t-1]})$, where $R_{[t_1,t_1],i} = \sum_{t=t_1}^{t_2} r_{t,i}$ and $C_{[t_1,t_1],i} = \sum_{t=t_1}^{t_2} |r_{t,i}|$ (note that the definition of $C$ is different than ours).

3. **Squint(.TV)** (Koolen & Van Erven, 2015): Squint is a parameter-free algorithm like AdaNormalHedge in that it can also be extended to priors over an initially unknown number of experts. Hence, we use the same idea as in AdaNormalHedge.TV (also see (Luo, 2017)) and apply the extension of the Squint algorithm for adaptive regret.

We used the (SCRATCH, 500, 200, 400) and (SCRATCH, 4000, 1000, 2000) configurations for the evaluations on the SVHN and CIFAR10 datasets, respectively.

### C.3 Results on Data-Starved Settings

Figure 6 shows the results of our additional evaluations on ImageNet and FashionMNIST in the *data-starved* setting where we begin with a very small (relatively) set of data points and can only query the labels of a small set of points at each time step. For both data sets, our approach outperforms competing ones in the various metrics considered – yielding up to 4% increase in test accuracy compared to the second-best performing method.

### C.4 Shifting Architectures

In this section, we consider the performance on FashionMNIST and SVHN when we change the network architectures from those used in the main body of the paper (Sec. 5). In particular, we conduct experiments on the *FashionNet* and *SVHNNet* architectures[11], convolutional neural networks that were used for benchmark evaluations in recent active learning work (Ash et al., 2019; Ash, 2021). Our goal is to evaluate whether the performance of our algorithm degrades significantly when we vary the model we use for active learning.

Fig. 7 depicts the results of our evaluations using the same training hyperparameters as FashionCNN for FashionNet, and similarly, those for SVHNCNN for SVHNNet (see Table 1). For both architectures, our algorithm uniformly outperforms the competing approaches in virtually all sample sizes and scenarios; our approach achieves up to 5% and 2% higher test accuracy than the second best-performing method on FashionMNIST and SVHN, respectively. The sole exception is the SVHN test loss, where we come second to CORESET – which performs surprisingly well on the test loss despite having uniformly lower test accuracy than OURS on SVHN (top right, Fig. 7). Interestingly, the relative performance of our algorithm is even better on the alternate architectures than on the models used in the main body (compare Fig. 7 to Fig. 2 of Sec. 5), where we performed only modestly better than competing approaches in comparison.

### C.5 Robustness Evaluations on Shifted Architecture

Having shown that the resiliency of our approach for both data sets for the configuration shown in Fig. 7, we next investigate whether we can also remain robust to varying active learning configurations on alternate architectures. To this end, we fix the FashionMNIST dataset, the FashionNet architecture, and the SCRATCH

---

[10]We also implemented and evaluated the method with uniform prior over time intervals, i.e., $p_{t,i} \propto \mathcal{I}_{t,i} \sum_{\tau=1}^{t} w(R_{[\tau,t-1],i}, C_{[\tau,t-1]})$ (without the prior $\frac{1}{\tau^2}$), but found that it performed worse than with the prior in practice. The same statement holds for the Squint algorithm.

[11]Publicly available implementation and details of the architectures (Ash et al., 2019; Ash, 2021): https://github.com/JordanAsh/badge/blob/master/model.py .

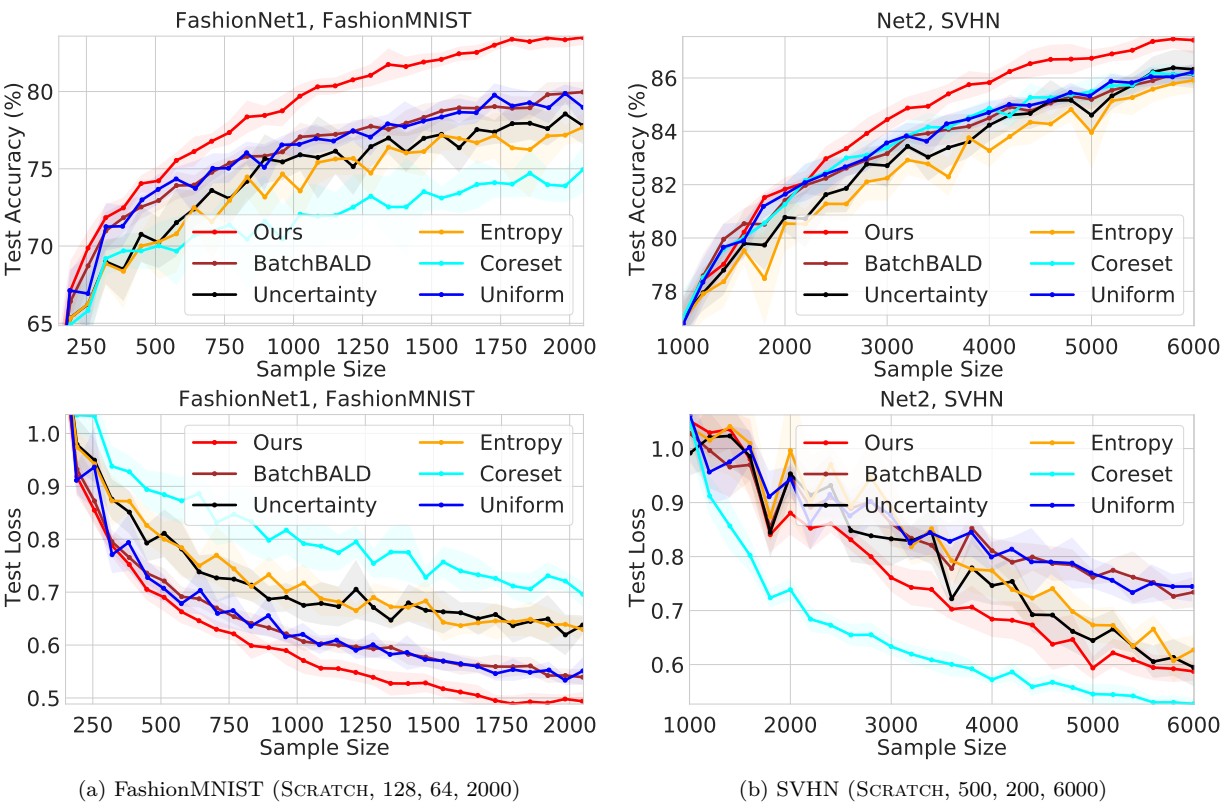

(a) FashionMNIST (SCRATCH, 128, 64, 2000)  (b) SVHN (SCRATCH, 500, 200, 6000)

Figure 7: Evaluations on the FashionNet and SVHNNet (Ash et al., 2019; Ash, 2021) architectures, which are different convolutional networks than those used in the main body of the paper (Sec. 5). Despite this architecture shift, our approach remains the overall top-performer on the evaluated data sets, even exceeding the relative performance of our approach on the previously used architectures.

option and consider varying the batch sizes and the initial and final number of labeled points. Most distinctly, we evaluated sample (active learning batch) sizes of 16, 96, and 224 points for varying sample budgets.

We present the results of our evaluations in Fig. 8, where each row corresponds to a differing configuration. For the first row of results corresponding to a batch size of 224, we see that we significantly (i.e., up to 3.5% increased test accuracy) outperform all compared methods for all sample sizes with respect to both test accuracy and loss. The same can be said for the second row of results corresponding to a batch size of 96, where we observe consistent improvements over prior work. For the smallest batch size 16 (last row of Fig. 8) and sampling budget (600), OURS still bests the compared methods, but the relative improvement is more modest (up to ≈ 1.5% improvement in test accuracy) than it was for larger batch sizes. We conjecture that this is due to the fact that the sampling budget (600) is significantly lower than in the first two scenarios (up to 6000); in this data-starved regime, even a small set of uniformly sampled points from FashionMNIST is likely to help training since the points in the small set of selected points will most likely be sufficiently distinct from one another.

## D  Discussion of Limitations & Future Work

In this paper we introduced ADAPROD$^+$, an optimistic algorithm for prediction with expert advice that was tailored to the active learning. Our comparisons showed that ADAPROD$^+$ fares better than GREEDY and competing algorithms for learning with prediction advice. Nevertheless, from an online learning lens, ADAPROD$^+$ can itself be improved so that it can be more widely applicable to active learning. For one, we currently require the losses to be bounded to the interval $[0, 1]$. This can be achieved by scaling the losses by their upper bound $\ell_{max}$ (as we did for the ENTROPY metric), however, this quantity $\ell_{max}$ may not be

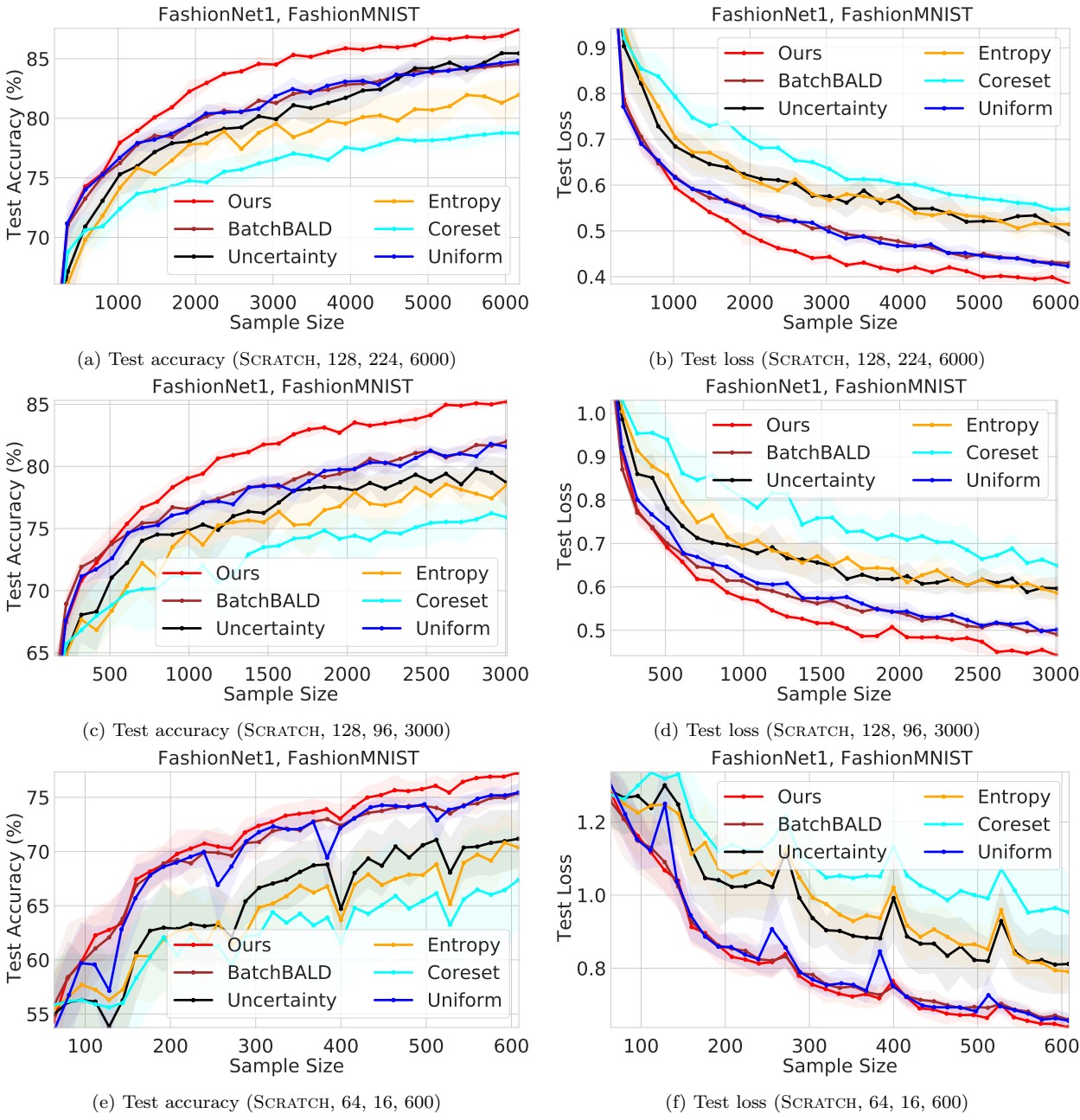

Figure 8: Evaluations with varying active learning configurations using the alternate FashionNet model trained on the FashionMNIST dataset.

available beforehand for all loss metrics. Ideally, we would want a scale-free algorithm that works with any loss to maximize the applicability of our approach.

In a similar vein, in future work we plan to extend the applicability of our framework to clustering-based active-learning, e.g., CORESET (Sener & Savarese, 2017) and BADGE (Ash et al., 2019), where it is more difficult to quantify what the loss should be for a given clustering. One idea could be to define the loss of an unlabeled point to be proportional to its distance – with respect to some metric – to the center of the cluster that the point belongs to (e.g., ≈ 0 loss for points near a center). However, it is not clear that the points near the cluster center should be prioritized over others as we may want to prioritize cluster outliers too. It is also not clear what the distance metric should be, as the Euclidean distance in the clustering space

may be ill-suited. In future work, we would like to explore these avenues and formulate losses capable of appropriately reflecting each point's importance with respect to a given clustering.

In light of the discussion above, we hope that this work can contribute to the development of better active learning algorithms that build on ADAPROD$^+$ and the techniques presented here.

