# OpenReview forum: "Low-regret Active Learning"
_TMLR — Rejected by TMLR_

### Review · Reviewer_wLDS · 2022-07-20

**Summary Of Contributions:**

This work studies the problem of active learning for deep networks for image classification problem. In active learning, at every data acquisition step, points to be labeled are chosen greedily based on some criterion evaluated using the model trained in the previous iteration. But, this trained model can be severely affected by the initial datapoints selected or the noise in the training process. Recently, deep ensembles have been used to evaluate the informativeness criterion to reduce the effects of training noise. But, training ensembles can be computationally expensive.

- This work formulates the labelling of points at each step as an online learning problem. Each datapoint can be considered as an expert in the online learning setup and the informativeness criterion of that point can be considered the gain at each step. Thus, selecting the point to be labeled can be seen as selecting the expert which maximizes the gain (or minimizes the loss) or minimizes the regret in the online setup. Using this framework, they are able to match the performance of using ensembles without needing to train them which can be computationally expensive.

- This work uses a previous algorithm for online learning with non-stationary stochastic environments with few changes and improvements. They also provide guarantees on the upper bound of the regret for their case. Their guarantee is a problem dependent guarantee which takes into account the hardness of the problem.

- This work provides comparisons with existing approaches like uniform sampling or greedy sampling without using an ensemble and show that they get improved performance over different configurations  and different datasets like CIFAR, FashionMNIST, SVHN and Imagenet.

**Broader Impact Concerns:**

This work does not have any ethical implications to the best of my knowledge.

**Requested Changes:**

- In the introduction, the authors write “Motivated by the widespread success of ensemble approaches in active learning (Beluch et al.,….”. It was hard to understand what exactly is the role of ensembles in active learning from this statement. The next line mentions training noise. It would be good to provide some more clarity on this like how deep networks are affected by noise like initialization or randomness in ordering of batches and how ensembles help in this regard. And, how this method is able to achieve this without training ensembles. At this point, it is not clear what is the connection of using regret framework with ensembles and noise. This would help the reader get an idea of what is going to come in the next part of the paper. This is mentioned in section 2.1 but a little more clarity here would be helpful for the reader.

- Again in section 2.2, explicitly mentioning how this method is alleviating the need to train ensembles would be helpful. Perhaps, making the connection to the setting of Wei et al. 2017 of how this environment is non-stationary and stochastic and we can think of training one model as a sample from infinite ensembles would be helpful.

- In algorithm 2, it would be good to mention the interpretation of $\eta$ (which is the learning rate) and $C$ if there is any.  It would make it easier for the reader to follow the algorithm. Some more intuition on the algorithm on how the weights are chosen in the particular way for example why increasing $C$ would reduce the learning rate would be helpful.

- On page 5 in generating predictions, $\hat{r}_t = \langle  p_t, \hat{l}_t \rangle - \hat{l}_t$. Here, $\hat{l}_t$ is used as both a $n$ dimensional vector and one dimensional vector. It would be good to clarify the notation here. Same in the notation section where $l_t(\theta\_{t-1})$ should be explicitly defined.

- Experiments on comparison with ensemble approaches would be good to include.

**Strengths And Weaknesses:**

The main strength of the paper is the idea of using online learning for active labelling of points which seems like a natural idea. This leads to computationally efficiency as well as performance improvement over several existing methods on a variety of benchmark datasets which is significant. Moreover, this method has provable regret guarantees.

Most of the previous works have focused on coming up with different informativeness criteria whereas this work focused on using those criteria and combine them with evaluating them on ensembles without the computational expense of training ensembles. So, this method can work on the top of all existing criteria.

The main weakness of the paper are that the writing of this paper is not clear. It is hard to understand the main contribution of this work and what this work is trying to achieve. I have outlined a few main points in the section below.

The experiments need to be more thorough. For example, I think the authors should compare their approach to the method which uses the ensemble and evaluate how well their approach can approximate the performance of this method.

The authors mention several sources from where advantage can come in like the presence of outliers or the noise in the training process. It would be good to include some ablation studies on how different sources of randomness in the training process like initialization, batch ordering changes the improvement. For the outliers or noisy points as well, it would be good to include some simulations on how exactly does this method help in this regard.

- In Lemma 1, why does the regret not go to 0 if t1 = t2?

- Is there a particular reason for calling the labels y as ORACLE(x)?

- This work mentions different sources of noise in training like data augmentation, initialization, dropout in section 2.1. Can the authors clarify what all randomness do they use when training models at different time steps?

- In section 3.2, can the authors give more information on the Optimistic Adapt-ML-Prod Algorithm on what exactly is the setting for which this algorithm is used?

---

> ### Author Response · Authors · 2022-08-10
> **Response to Reviewer wLDS**
>
> Thank you for your careful consideration of our paper and helpful suggestions to clarify the contributions of our work. Please find below our respective responses to the questions raised in the Strengths And Weaknesses and Requested Changes sections.
>
> #### Strengths And Weaknesses
>
> 1. You are right that the regret should be 0 for $t_1 = t_2$. The log factors are an artifact of the analysis and the upper bounds we use in the proof. We will clarify the lemma statement to instead be $t_1 < t_2$ to make the statement more precise.
>
> 2. We used the ORACLE(x) call to make it explicit that we are having to query the label of the point and that the labels are not available in advance. We are happy to change this notation to just $y$ if the reviewer thinks it would help improve the exposition.
>
> 3. The full details of the settings used for our experiments can be found in Sec. C of the Appendix. We include the relevant parts here for completeness. The randomness in the model initialization, the stochastic optimizer (Adam or SGD) in selecting mini-batches, and random data augmentation (random cropping and horizontal flips) are the three sources of randomness present in all our experiments. Please see Sec. C.1 for the full details. We are happy to move over some of this text to the main paper to make the sources of randomness explicit.
>
> 4. Optimistic-Adapt-ML-Prod [1] can be used for making decisions in both non-stationary adversarial and stochastic environments (Theorems 3.4 and 3.5 of [1]). We will clarify the exact setting (stochastic, as you correctly pointed out) for which OAMLProd is used in our revision.
>
> #### Questions in Requested Changes
> 1. Thank you for your constructive feedback and help in clarifying the sources of randomness that we consider and how ensembles help mitigate noise. We will write an abridged version of the discussion surrounding this point in Sec. 2.1 in the introduction as well so that it is more clear. In our revision, we will also mention the connection to the setting of Wei et al. as you helpfully suggested (below).
>
> 2. You are fully correct on the connection between the setting of Wei et al. and the active learning setting we consider. Exactly as you pointed out, the loss vector $\ell$ that is a random variable with respect to the trained model can be viewed as the loss being a stochastic loss that is a sample from a stochastic environment.
>
> 3. The learning rate $\eta$, analogous to the learning rate in Gradient Descent (GD) algorithms, controls how aggressively we update the probabilities (akin to the weights of GD) at each time step. In the online learning setting that we consider, this means that if $\eta_{t,i}$ is low, we make more conservative (smaller) updates to the probability of sampling $i$, and vice versa. Hence, it makes sense to update with a low value of $\eta$ if there is too much variability in the observed losses or uncertainty surrounding expert $i$. With this interpretation, $C$ plays the role of measuring the amount of variations or unpredictability in the predicted losses ($\hat \ell_t$ which defines $\hat r_t$) and the actual losses ($\ell_t$ which defines $r_t$). If $C_{t,i}$ is high, then the losses of expert $i$ are not very predictable, which means we should be more conservative in updating the probability $p_{t,i}$ and hence should use a smaller learning rate instead. Thank you for the helpful suggestion – we will include a summary of this discussion in the paper.
>
> 4. You are correct that $\hat r_t = \langle p_t, \hat \ell_t \rangle - \hat \ell_t$ is a slight abuse of notation (also in the definition of $r_t$) since $\langle p_t, \hat \ell_t \rangle$ is a scalar and $\hat \ell_t$ is an $n$-dimensional vector. The proper way to define it would be to $\hat r_t = \langle p_t, \hat \ell_t\rangle  1_n - \hat \ell_t$, where $1_n$ is an n-dimensional vector of all $1$s. We will revise the old definitions to use this new one everywhere that is relevant in the paper.
>
> 5. For the sake of fair comparisons, we allowed a single model to be trained by each algorithm in our evaluations. We are happy to report previously presented empirical results that demonstrate the power of ensembles in active learning and the relative accuracy improvements on popular vision data sets (up to 10% improved test accuracy on the same vision data sets we considered).
>
> [1] https://arxiv.org/pdf/1712.00578.pdf
>
> [2] The Power of Ensembles in Active Learning https://openaccess.thecvf.com/content_cvpr_2018/papers/Beluch_The_Power_of_CVPR_2018_paper.pdf

---

### Review · Reviewer_3vu1 · 2022-07-27

**Summary Of Contributions:**

This paper formulates the selection of examples to label as prediction  with sleepy expert problem, where an indicator function is introduced to make sure the same example is not selected twice. The algorithm is an extension of the Optimistic Adapt-ML-prod algorithm (Wei et al, 2017). Both adaptive regret and adaptive regret bounds are developed, which involve drift and variance in the upper bounds. Experimental results on four vision datasets are also presented to show the effectiveness of the proposed algorithm.

**Broader Impact Concerns:**

No worries on impact concerns.

**Requested Changes:**

Below Theorem 2, the authors mention that their regret bounds can be sublinear in practice. It would be interesting if the authors can provide a specific example where both V_T and D_T can be estimated and the regret bounds are sublinear.

In Section 2, the authors mention the uniform sampling as a baseline. What would the regret bounds be if we choose the uniform sampling strategy? Can the proposed methods enjoy clear advantage as compared to the uniform sampling strategy in terms of regret bounds?

It is not clear to me the meaning of controlling $\sum_{t=t_1}^{t_2}r_{t,i}$ in Lemma 1. Note you use the same $i$ here. However, in active learning we should choose an example at most once. Therefore, it is not quite meaningful if we compare a strategy choosing the same examples in all rounds. It would be helpful if the authors can provide some explanations.

The title "Results" in Section 5 can be replaced by "Experimental Results"

Minor comments:
Section 2.2: ", The"
Section 2.2: "picking a points"
Section 4.2: "S^*_{tj}\in[n]" should be "S^*_{tj}\subset[n]"
Section 5: "our codebase be found"

**Strengths And Weaknesses:**

Strength:
The study of active learning from the perspective of prediction with experts is interesting. The algorithm and the analysis introduce several techniques to adapt to the active learning setting where each example is only selected once. The paper is very well written. The theoretical analysis and experimental analysis are convincing from my understanding.

Weakness:
The paper presents regret bounds in terms of V_T and D_T. However, the paper does not give practical examples on how these terms would grow in practice. The comparison with uniform sampling is also not clear to me.

---

> ### Author Response · Authors · 2022-08-10
> **Response to Reviewer 3vu1**
>
> Thank you for your constructive and detailed feedback. Our responses to the concerns raised in the requested changes are below. We will include the discussion below in our revision.
>
> 1. Regarding an example with $\mathcal V_T$ and $\mathcal D_T$: this is a great suggestion and we will dedicate a paragraph in the main body of the paper that explains this point. For the sake of completeness in this discussion, we wanted to sketch the bounds for a simple example. Consider the case where the losses are stationary (fixed over all $t$) and do not depend on the previous samples or the noise, i.e., $\ell_t = \ell$ for all $t \in [T]$. In this case, using $\hat \ell_t = \ell_{t-1}$ as our predictions, $\mathcal V_T = O(1)$ since there is no variation in the losses. Moreover, since the losses are constant, the probabilities $p_t$ do not change significantly after a certain number of rounds. This implies that $E[r_t] \approx E[r_{t-1}]$ for $t \ge t_0$ for some $t_0$, which in turn leads to $\mathcal D_T = o(T)$. Plugging these values for the expressions $\mathcal D_T$ and $\mathcal V_T$ in the regret bound of Theorem 2 culminates in a sublinear regret bound.
>
> 	To see how uniform sampling fares, consider the stationary loss vector $\ell$ to be $\ell = (1, 1, 0…, 0)$ where $\ell_i = 1$ for $i \leq k$ and $\ell_i = 0$ for all $i > k$ for some appropriately chosen $k$. Clearly, it would be best to not allocate any probability of sampling to the first k points, and always pick from the remaining points with indices $i > k$. However, since uniform sampling always allocates a uniform portion of the probability mass on the first $k$ points, and since for an appropriate $k$, at least some of the $k$ points do not get chosen for a significant number of rounds $c T$ for $c \in (0,1)$, uniform sampling incurs linear regret. We will include a detailed version of this example and surrounding discussion in the Appendix. Thank you for your helpful suggestions to make our bounds practically relatable.
>
> 2. $r_{t,i}$ is the instantaneous regret with respect to point i and is defined as $r_{t,i} = \langle p_t, \ell_{t}\rangle - \ell_{t,i}$. In words, this measures the performance gap between the expected performance of sampling according to $p_t$, $\langle p_t, \ell_{t}\rangle $, and the performance of an approach that would deterministically pick point $i$ by putting all of the probability mass on $i$, $\ell_{t,i}$. Since the entire loss vector $\ell_t$ is visible in each active learning iteration (as described Sec. 2.2), we can measure the instantaneous regret $r_{t,i}$ for all $i \in [n]$ -- even for points that we do not directly sample.
>
> 3. Thank you for your suggestions regarding the title and typographical errors. We have implemented all recommended changes.

---

### Review · Reviewer_FVca · 2022-07-28

**Summary Of Contributions:**

This paper considers the problem of active learning: choosing informative points to label in a sequential fashion to learn a classifier that is as good as one trained on the full dataset. To do so, the authors rely on the scheme of information gain maximization, which greedily labels points based which maximize an acquisition function. The authors argue that in general, greedy methods are sub-optimal, and instead provide a methodology based on online learning. Their method maintains a distribution over the points, and uses an online learner to update this distribution. The underlying losses for the online learner come from the gain functions of previously learned models. Their online learning algorithm is based on the Adapt-ML-Prod algorithm of Wei et. Al ’17, whose regret bound scales with the cumulative variance and $\ell_\infty$ deviations of the observed loss functions.




**Broader Impact Concerns:**

None.

**Requested Changes:**

To secure my recommendation, I would need the concerns above, especially those concerning the online algorithm addressed.

**Strengths And Weaknesses:**

Overall I thought the paper was well written and did a good job outlining the problem.

Here are some concrete questions/comments:

- What do you mean by training noise? Are you assuming noisy labels? Noise in a training procedure? It should be clarified further. Also, it’s not clear why the loss should be a function of  $\xi_{t-1}$ if it depends on model $\theta_{t-1}$.

- The definition of loss is a bit strange - in general in classification settings, when we talk about loss, we mean the loss in terms of prediction error or some surrogate (i.e. logistic loss). Loss in this paper is 1-gain, where the gain is related to the uncertainty of choosing a point.

- This is related to my main concern: Why the online learning approach?  From the discussion in 3.5, it seems that the strategy is to compute $p_t$, get the losses with respect to $\ell_t$, and then iterate. However, given model $\theta_{t-1}$ - can’t I just compute the gain from each point and choose greedily? This would achieve zero regret under the current metric. Thus it seems like the online algorithm is being used to inject some additional exploration to help with the active learning. This is fine, but perhaps there are more principled ways to do this than modeling this as an online game. It’s very likely I have missed something. Can the authors clarify if my understanding is correct?

- I’m not sure what the regret analysis really tells us. Again, in the end of the day in classification, we care about the classification accuracy. I am not sure what a bound on the regret really tells us about the performance of the active learning algorithm for minimizing classification error.

The experiments consider a variety of standard datasets and architectures. The authors show a modest active gain with less computational effort. The active gain does not seem as clear for CIFAR10 with oney 10 repetitions (for example if 95% CI’s were used instead of one standard error).  Figure 4 is particularly interesting - it shows that the ADAPROD+ demonstrates a gain regardless of the underlying acquisition function. The paper also has several robustness checks that vary the underlying architectures and acquisition functions that lend credibility to the experiments. My only concern about the experiments is that comparison to BADGE (which is cited) seems to be missing.

Summary: In the abstract, the authors say that their main goal is to gain the wins of ensemble methods without the computational burden of using ensemble methods. To do so they employ an online learning method. Empirically this method shows promises, but I struggle to understand the theoretical underpinnings which make up most of the paper.

---

> ### Author Response · Authors · 2022-08-10
> **Response to Reviewer FVca**
>
> Thank you for your careful consideration of our paper and for your helpful feedback. Please see our respective responses to the questions raised in the Strengths and Weaknesses section below. We will include the clarifications and discussions below in our revision.
>
> 1. The training noise here does not refer to label noise. Instead, it refers to the various sources of randomness encountered during the training process: (i) random network initialization, (ii) randomness in SGD, or more generally, the selection of batches for the gradient-based optimizer, (iii) Dropout, and/or (iv) randomized data augmentation (e.g., random crops/translations of images). As we describe in Sec. 2.2 (paragraph Notation), we model this randomness by the random variable $\xi_{t-1}$. We apologize for the confusion regarding $\xi_{t-1}$ and $\theta_{t-1}$. You are correct that we are interested in the expected loss over the random $\theta_{t-1}$, but conditioning on our past actions and observations, this simplifies to just considering the randomness over $\xi_{t-1}$ since $\theta_{t-1}$ is a deterministic function of $\xi_{t-1}$ (page 4, second-to-last sentence of paragraph 1).
>
> 2. The loss is a function of a user-defined proxy for the informativeness of each point: points with high losses are (generally) not beneficial for training, whereas those with low losses are. In the paper, we used 1 - gain as an example instantiation of our framework, where gain was some measure of the uncertainty over point $x_i$. Nevertheless, our method is general enough to be used with any bounded user-specified notion of gain and loss as we outline in Sec. 5.4. For example, in Sec. 5.4, we define the gains (and in turn, the losses) using the BALD, Entropy, and Uncertainty metrics. Our results in Fig. 4 show that AdaProd+ instantiated with these gains improves upon the greedy variants for each of these measures. We will clarify in our revision that the loss here is not the training loss that is popularly referenced in ML.
>
> 3. For a given model $\theta_{t-1}$, you are right that the *model-specific* loss of the greedy approach with respect to $\theta_{t-1}$, $\ell(\theta_{t-1})$, would be 0. However, we are not interested in achieving low regret with respect to particular observations of $\theta_{t-1}$, but rather interested in minimizing the loss over the randomness in the training, i.e., $\mathbb E_{\xi_{t-1}}[\ell_t(\xi_{t-1})]$. To see the rationale for this formulation, consider what would happen if we were to get unlucky and the training diverges. This would mean that $\theta_{t-1}$ provides little to no information whatsoever about the importance of each point. If we were to use the greedy approach, it would rely entirely on $\theta_{t-1}$ to define the losses, and lead to a misguided selection of points.
> A more robust approach would be to train an ensemble of network models – to smoothen out the training noise by approximating $\mathbb E_{\xi_{t-1}}[\ell_t(\xi_{t-1})]$ via the sample mean over the ensemble – and then apply the greedy approach over $\mathbb E_{\xi_{t-1}}[\ell_t(\xi_{t-1})]$. This smoothening leads to significant performance gains in practice [1] and motivates our regret formulation. However, this method is computationally costly as it requires a large ensemble of networks to approximate the expectation. This is precisely where our regret formulation comes in. We compare the performance of our approach (which uses a single trained network) to that of the greedy approach that has access to an ensemble of infinite size from which it can compute $\mathbb E_{\xi_{t-1}}[\ell_t(\xi_{t-1})]$. By minimizing the regret with respect to the *expected loss over the randomness in the training*, we are minimizing the performance gap between our approach and that of the greedy approach with an omnipotent ensemble.
>
>
> 5. Low regret in this context implies that we can be provably competitive with an ensemble method without the computational burden of having to train an ensemble (as stated in the Abstract, Introduction, and Problem Definition sections). You are fully correct that this does not necessarily prove that we will have a higher test accuracy, however, this is a limitation of virtually every active learning approach for deep learning: a proxy is used to define the informativeness of each point. For example, margin sampling and entropy sampling rely on the assumption that labeling points that the model is most uncertain about would be most helpful for achieving a higher test accuracy. Besides, our framework can be instantiated with any definition of losses/gains. This means that the performance of AdaProd+ scales with future developments in defining the losses in a way that better correlates with the classification error. We will make this explicitly clear in our revision.
>
>
> [1] https://openaccess.thecvf.com/content_cvpr_2018/papers/Beluch_The_Power_of_CVPR_2018_paper.pdf

---

### Decision · Action_Editors · 2022-09-10

**Recommendation:** Reject

**Comment:**

After the discussion period, the reviewers recommend that:

- the paper adds additional experimental results comparing to ensemble-based active learning. In Beluch et al 2018, even an ensemble of size 3-5 seems to provide a significant gain, so it is an important baseline to compare with. Specifically, the authors are encouraged to: (1) add the extra plots on the performance of ensemble-based method, as mentioned in author feedback; (2) compare their methods with a relatively computationally-light ensemble-based baseline with ensemble size of, say, 3-5, both in terms of running time and label query complexity.

- revise the paper to address the clarity issues raised by the reviewers.

Based on the above, we recommend a "major revision", and encourage the authors to resubmit a revised manuscript as a new submission. (the authors will then be asked to include a link to this previous submission.)